# Can green funds improve corporate environmental, social, and governance performance? Evidence from Chinese-listed companies

**Fangjun Wang[1], Xinmiao Zhou[1], Tian Gan[2]***

**1** School of Business, Ningbo University, Ningbo, China, **2** School of Business, East China University of Science and Technology, Shanghai, China

\* gantian13917167084@163.com

**Data Availability Statement:** All relevant data are within the manuscript and its Supporting information files.

## Abstract

Green funds play pivotal roles in driving corporate sustainable development. Utilizing data from Chinese publicly listed companies from 2010 to 2021, we examine the impact of green funds on corporate environmental, social, and governance (ESG) performance and the underlying mechanisms. The research findings claim that green funds positively affect corporate ESG performance. Mechanism analysis systematically demonstrates that green funds contribute to elevated corporate ESG performance by alleviating financial constraints, enhancing managerial efficiency, and fostering green innovation. Heterogeneity analysis further underscores that the effect of green funds is particularly potent in companies with high external attention. Furthermore, green funds also play significant roles in production capabilities and economic value. This research enriches the micro-level evidence on the development of green funds and furnishes substantial implications for sustainable development.

## Section 1: Introduction

In recent years, the escalating global concern for climate change and environmental issues has prompted an increased focus on ESG matters and sustainable development [1]. In 1992, the United Nations Environment Assembly urged financial institutions to integrate ESG factors into their investment decisions. Similarly, the European Union implemented the Non-Financial Reporting Directive in 2014, firmly embedding ESG regulations into policy frameworks. Additionally, the Chinese government has established ambitious targets to peak carbon emissions by 2030 and achieve carbon neutrality by 2060, highlighting the need to shift towards a more environmentally friendly development trajectory for promoting sustainable and low-carbon socio-economic progress. These strategic initiatives represent pivotal measures towards transitioning to higher quality and more sustainable development. Moreover, the China Securities Regulatory Commission (CSRC) is considering the inclusion of mandatory ESG disclosures in regular reporting by listed companies, guiding the improvement of corporate ESG performance [2, 3].

**Funding:** This work was supported by the National Social Science Fund of China [21BJY002], National Natural Science Foundation of China [72373078], the National Social Science Fund of China [20AZD033], Shanghai Planning Office of Philosophy and Social Science [2023EJB005], Shanghai Post-doctoral Excellence Program [2023237]. The funders had no role in study design, data collection and analysis, decision to publish, or preparation of the manuscript.

**Competing interests:** The authors have declared that no competing interests exist.

Given the positive effects of ESG on business performance [4], corporate value [2, 5], corporate financing [6], and corporate innovation [7], ESG investments are gaining traction among companies [8]. Consequently, the significance of ESG principles in corporate production and operational decision-making continues to rise. While some researches have tested the influence of environmentally oriented financial tools in augmenting corporate ESG performance [9, 10], limited attention has been given to the impact of green funds on ESG performance. As green funds become integral parts of green finance, they play essential roles in green transformation and sustainable development. Therefore, a comprehensive exploration of the effect of green funds on corporate ESG performance represents an imperative avenue of research and serves as a central focus of this study.

Green funds, as a distinct category of institutional funds, encompass fund entities in the capital market that integrate economic, environmental, and social responsibility [11] since they evaluate corporate financial performance and emphasize a range of social and environmental impacts. Moreover, they exercise rigorous oversight and governance over companies, prompting firms to actively embrace corresponding social responsibility alongside their pursuit of economic interests, and improving the dual effects of expanding financial returns and environmental quality [12]. Like green bonds, green funds provide an external avenue for corporations to access green financing. Since green funds focus more on environmental responsibility goals, the investment decisions pay more attention to environmental standards, pollution control effects, and ecological protection [13, 14]. A recent study strongly recommended that green funds benefit market performance [15], stimulate environmental investments [11], and encourage pollution-abating measures [16]. However, there is limited comprehensive research focusing on the impact of green funds on ESG performance.

The question arises: Do green funds positively influence corporate ESG performance? What mechanisms underlie this impact? Do these effects differ based on varying external attention levels and firm or regional attributes? Furthermore, while enhancing corporate ESG performance, do green funds positively affect production capabilities and economic performance?

Our study seeks to answer the previously mentioned questions by systematically investigating how green funds affect corporate ESG performance. To identify the impact of green funds and the underlying mechanisms, we establish a fixed-effects model with the sample of listed firms on the Shanghai and Shenzhen Stock Exchange from 2010 to 2021. According to our findings, green funds positively influence corporate ESG performance. The impact on corporate social responsibility is relatively weaker compared to that on environmental and governance performance. To address potential endogeneity concerns, we employ instrumental variable methods and the Heckman two-step verification approach, whose results are consistent with the baseline regression. Furthermore, a series of tests, including replacement of key variables, considering the lagged effect and other important factors, alternative model specifications, placebo tests, and Propensity Score Matching (PSM), have been conducted to ascertain the robustness of our findings. In terms of the underlying mechanisms, green funds reduce the financing costs for firms and augment internal fund availability, thereby alleviating the overall financial constraints of firms. Then, green funds elevate managerial efficiency by improving internal control quality, reducing the myopia of management, and promoting the total asset turnover rate to improve corporate ESG performance. The involvement of green funds enhances the green innovation capability and contributes to the advancement of ESG performance. An analysis of heterogeneity reveals that companies with higher external attention experience a more pronounced enhancement in ESG performance due to the involvement of green funds.

Regarding corporate characteristics, non-SOE, heavily polluting, and large-scale enterprises exhibit a more pronounced improvement in ESG performance through the engagement of green funds. Consequently, it may contribute to an amplification of differences among enterprises, creating environmental sustainability gaps between more and less resourceful entities. Additionally, concerning regional characteristics, the influence of green funds on corporate ESG performance is particularly noticeable in companies located in regions characterized by lax environmental regulations, lower levels of marketization, and higher air pollution. Notably, green funds promote companies' total factor productivity and market value. The findings of this study provide theoretical and specific empirical support for the value of green funds for the sustainable development of enterprises, further enhancing our comprehension of the consequences of green funds on businesses and propelling the advancement of ESG practices in Chinese enterprises.

The study generates several potential contributions. Firstly, it expands the research on the determinants of corporate ESG performance. While previous scholars have explored some factors influencing corporate ESG performance from the perspective of digital financial development [17], digital technological innovation [18], corporate digital transformation [19], managerial attributes [20, 21], corporate governance structures [22, 23], and government environmental attention [24–26], this study aims to investigate the impact of green funds on corporate ESG performance and adds to the factors contributing to corporate ESG performance. Furthermore, it reveals how green funds impact corporate ESG performance through three channels: financial constraints, managerial efficiency, and green innovation. Our study not only adds to the theoretical understanding of corporate sustainable development but also offers practical insights that can inform strategies for fostering ESG performance.

Secondly, a substantial body of literature has widely discussed the effect of green finance on corporate performance. Green finance offers various financial services for firms engaged in sustainable green projects and exerts external constraints to regulate the use of green funds and the environmental protection behavior of enterprises [3]. As for economic performance, prior studies discovered that green finance can lower debt financing costs [27], increase R&D investments [28], inhibit corporate over-investment behavior [29], enhance total factor productivity [30, 31], and bolster market value [32]. As for environmental performance, green finance can stimulate green innovation [33, 34], deter greenwashing behavior [3], reduce energy consumption intensity [35], curtail carbon emissions [36], and facilitate the green transformation of heavily polluting enterprises [37, 38]. However, the previous research mainly investigated the effects of green loans and bonds on micro-level corporate behavior. As an essential branch of green finance, green funds are bound to positively impact enterprises' sustainable development. To overcome these listed situations, this article launches an experimental exploration of how green funds impact corporate ESG performance. The results would better prompt enterprises to recognize the significance of green funds.

Lastly, this study considers the heterogeneous effects of green funds on corporate ESG performance from a micro-level perspective, broadening the analysis to encompass how external scrutiny impacts corporate ESG performance. External attention is primarily characterized by external pressures rather than incentives [39]. The study reveals notable variations in the influence of external attention from media, analysts, and the general public [39–41], extending the body of research on how external examination influences corporate ESG practices and providing valuable evidence on how green funds effectively guide companies to enhance their ESG performance.

The remainder of the paper is structured as follows: **Section 2** introduces the development of green funds in China, presents the theoretical foundations, and outlines our research hypotheses. **Section 3** provides details on the data sources, variable selection, and the

construction of the baseline model. **Section 4** encompasses the empirical findings, including validity tests of green funds, baseline regression analysis, tests for endogeneity, robustness checks. **Section 5** provides the mechanism analysis and exploration of heterogeneity. **Section 6** makes the further discussion. **Section 7** concludes this study, offers the policy implications, and discusses the generalizability and limitations.

## Section 2: Background, theoretical analysis and hypotheses development

### 2.1 Development of green funds in China

Green funds play vital roles in directing corporate attention towards their own environmental and social responsibility performance and providing financial support for ESG initiatives. The China Securities Investment Fund Association (CSIFA) introduced the "Guidelines for Green Investment" to conceptualize the scope of green investment, guide investors in cultivating eco-friendly principles, and enhance the environmental outcomes of investment activities for establishing green financial and investment mechanisms. As a significant form of green investment, green funds are proliferating by policy initiatives and representing a heightened focus on the ESG performance of investment projects compared to conventional funds, thereby fostering sustainable development.

Fig 1 depicts the changing trend in the number of firms with green funds and their proportion relative to the total number of listed companies in China each year. Notably, during 2010–2016, the number of firms with green funds displays a general upward trajectory. Although there is a minor decline in the count of such enterprises during 2016–2019 due to the exit of some green funds, the number surges again from 2019 onwards, peaking in 2021.

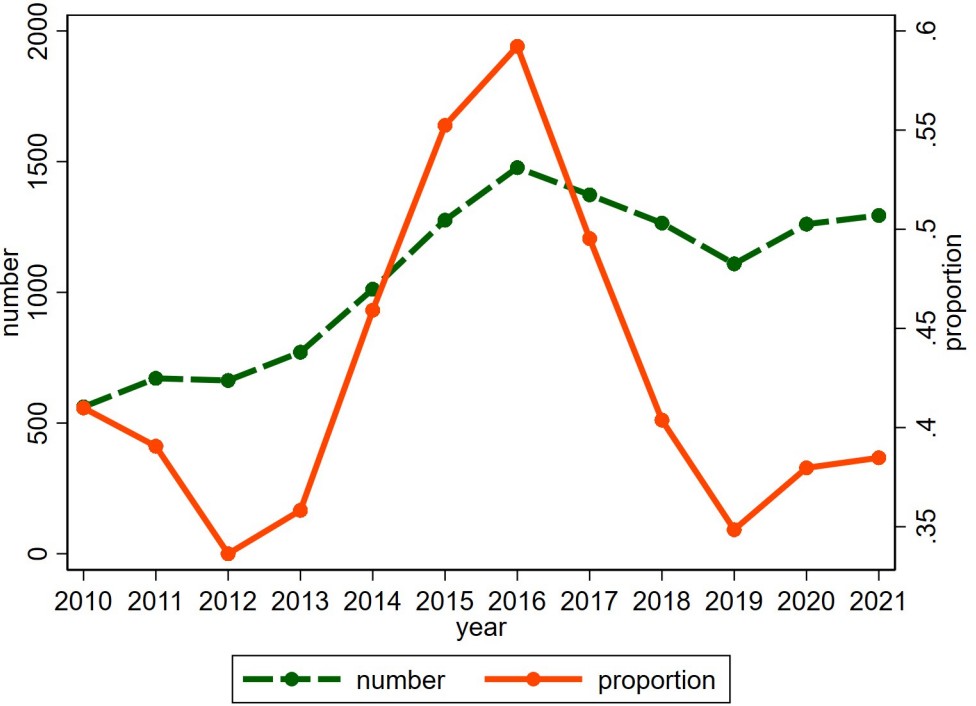

**Fig 1. The number and proportion of green fund companies.**

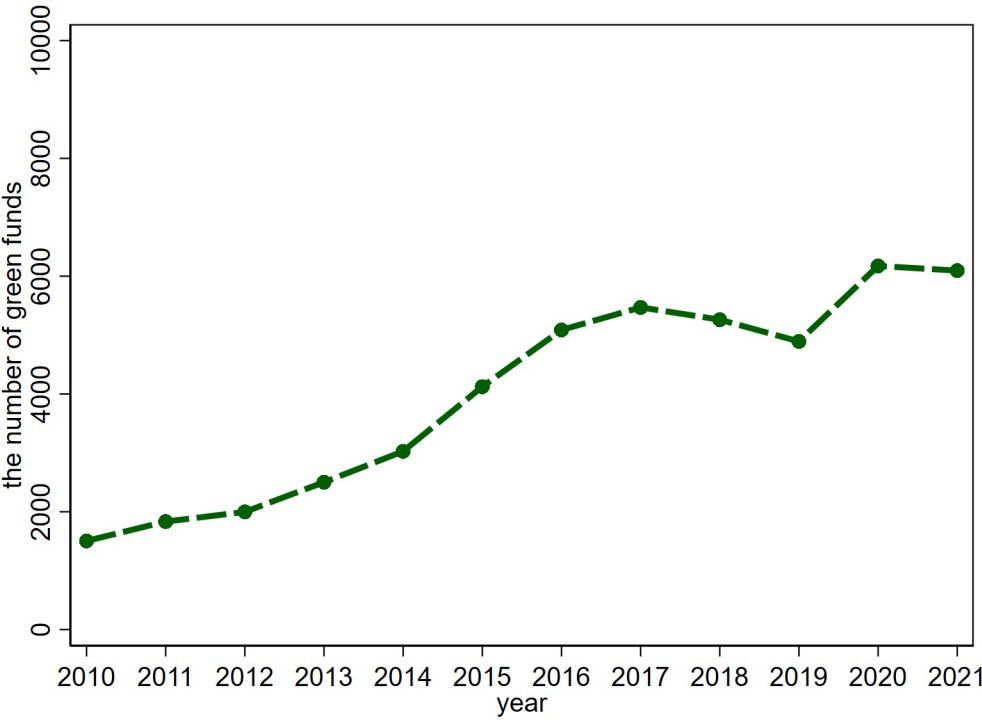

**Fig 2. The number of green funds.**

Except for 2010–2012 and 2016–2019, the proportion of enterprises with green funds exhibits an upward trajectory throughout the sample period.

Fig 2 illustrates the annual changes in the total count of green funds in China. Apart from a slight decrease observed during 2017–2019, the number of green funds steadily increases in the remaining years of the sample interval, signifying an expanding scale of green funds in China.

## 2.2 Theoretical analysis and hypotheses development

**2.2.1 Green funds and corporate ESG performance.** Stakeholder theory asserts that firms can optimize their value by addressing the demands of their stakeholders to facilitate development [42]. Given the diverse interests of these stakeholders, firms must consider numerous factors during their decision-making processes [43]. Green funds assist firms in prioritizing their ESG performance, cultivating a more favorable public perception and ultimately augmenting their intrinsic value. Hence, aside from profit maximization, firms should also conscientiously emphasize their social and environmental responsibility as significant contributors to their market value. Consequently, companies should also thoughtfully acknowledge their social and environmental responsibility, which contributes to long-term and sustainable investments in the market. Although the external pressure exerted by green funds might momentarily hinder the immediate attainment of substantial profits, it indicates that embracing social and environmental responsibility can lead to sustainable profits from a long-term perspective [44].

Furthermore, green funds inherently possess a green attribute, signifying that they must be directed towards projects aligned with environmental conservation and social responsibility,

precluding their diversion for alternative uses. The infusion of green capital provides a safe-guard for the implementation of ESG initiatives. Lastly, the value stemming from corporate ESG performance can be shared with stakeholders, fostering improved stakeholder relation-ships and collaborative support for sustainable development, which mitigates misleading or deceptive actions towards government regulatory bodies, the wider public, and other stake-holders. Such an approach is congruent with green funds' objectives and assists firms in elevat-ing their ESG performance. Hence, this study advances the following hypotheses:

*Hypothesis 1*: Green funds can improve corporate ESG performance.

**2.2.2 Green funds, financial constraints, and corporate ESG performance.** Green finance allows enterprises to undertake green projects by restricting funding to heavily pollut-ing and energy-intensive companies, and alleviating financial constraints in green investment projects by reducing financing costs and improving financial structures. Existing literature mentions that green credit policies lead to varying credit scales and terms for firms of different pollution levels, resulting in decreased financial costs for environmentally conscious firms, and increased financing costs for heavily polluting companies [38]. Notably, the increase in financing costs for heavily polluting firms remains significant over extended time horizons [45]. Green bonds also contribute to optimizing corporate financing conditions by increasing long-term financing proportions and reducing corporate debt financing costs [10].

The implementation of corporate ESG performance requires substantial funding through continuous financing activities. In this context, green funds infuse the requisite capital directly into corporate green endeavors while enhancing access to financing opportunities and reduc-ing financial costs. Additionally, guided by signal theory, firms with green funds convey posi-tive signals of their ongoing or impending engagement in green transformation. As external investors increasingly prefer financing aligned with ESG standards [8], releasing these green signals benefits the reduction of financing thresholds. Given that debt and equity financing constitute the primary financing sources for companies, the decrease in financing thresholds corresponds to fallen financing costs for both sources, ultimately improving the overall financ-ing status and facilitating a greater flow of internal cash for ESG initiatives.

Due to the resource-intensive and extended lifecycle nature of green projects, consistent and long-term capital becomes essential for firms engaging in eco-friendly production [10]. Limited financing hampers the realization of environmental projects, diminishing enthusiasm for green research and development [46] and inadvertently leading to increased pollution emissions [47]. With the support of green funds, enterprises can effectively participate in envi-ronmental conservation [48] and social responsibility activities [49], thereby enhancing satis-faction among internal employees and external overseers. Moreover, from an investor perspective, firms demonstrating strong environmental and social responsibility acquire financing at a lower cost because investors prefer sustainable practices [50], even if immediate returns might not be substantial [51]. Guided by investor preferences for sustainable invest-ment, companies greatly enhance ESG performance. Based on these insights, Hypothesis 2 is proposed in this study:

*Hypothesis 2*: Green funds can alleviate corporate financial constraints, thereby enhancing corporate ESG performance.

**2.2.3 Green funds, managerial efficiency, and corporate ESG performance.** As vital components of external investment, green funds play crucial roles in conducting adequate external supervision of companies, mitigating agency problems, and promoting managerial

efficiency. The separation of ownership and control often creates information asymmetry between shareholders and managers [52], causing managerial actions that may not align with the long-term interests of shareholders and giving rise to agency costs. Green funds can alleviate information asymmetry between external investors and corporate managers. In contrast to typical institutional investors, green fund investors possess specialized and comprehensive environmental knowledge and green governance experience [11]. They can participate in relevant decision-making processes and establish various communication channels with managers. These approaches allow them to understand the true state of production operations and their fulfillment of environmental and social responsibility, thereby determining whether the current development of the company aligns with the initial intention of green sustainable transformation.

Elevating internal control within a company is beneficial for enhancing information transparency and releasing agency concerns [53]. According to activist shareholder theory, green fund investors can exert effective external oversight, reinforce the supervision and constraint of managerial opportunistic behavior, and enhance the effectiveness of internal controls. Specifically, green fund investors can articulate their expectations for sustainable development through various ways, such as submitting shareholder proposals, collaborating with other shareholders, engaging in private communication, and negotiating with management to further weaken the information asymmetry between corporate managers and green fund investors, foster a friendly environment for information exchange within the company and improve internal control systems.

Corporate governance structures are optimized by reducing agency costs, raising managerial sustainability awareness, and improving internal control quality [10]. These empower companies to effectively implement ESG practices while attaining their established production and operational objectives. Guided by the concept of sustainable development, ESG practices harmonize decision-making and implementation, ultimately enhancing economic, social, and environmental benefits. Building on this evidence, Hypothesis 3 is proposed in this study:

***Hypothesis 3***: Green funds have the potential to enhance corporate ESG performance by elevating managerial efficiency.

**2.2.4 Green funds, green innovation, and corporate ESG performance.** Green finance facilitates green innovation in corporations, improving environmental performance [33, 46, 54]. By providing financial support and allocating diverse financial resources, green finance equips enterprises with supplementary capital for improving green technologies and pioneering environmentally sustainable projects [46], which sparks the drive for engaging in green innovation endeavors in turn. Moreover, green finance facilitates accessible financing for green projects and imposes stringent controls on investments in pollution-intensive ventures, significantly constraining funding levels for heavily polluting and highly energy-consuming firms. These force companies to carry out green innovation. By supporting green projects and technologies, green finance contributes to improved environmental quality and reduced energy consumption, ultimately leading to the achievement of green transformation [35]. In essence, green innovation serves as a vital channel for green funds to exert the environmental governance effect.

Based on the innovation compensation theory, firms can only elevate their environmental performance, establish green reputations, and attract the interest of green funds by prioritizing investment in green technology research, improving the efficiency of clean energy utilization, and phasing out outdated capacities [55]. Green funds encourage firms to promote green innovation [56]. Furthermore, compared to conventional innovation, green innovation often

involves higher technological complexity and uncertainty. Firms grapple with prolonged research cycles, limited funding available, and a lack of incentives for innovation, hindering the motivation for green innovation. Green fund investors enhance fund allocation efficiency for green innovation activities and leverage their environmental knowledge and understanding of advanced technologies to transmit cutting-edge green insights and experiences, thereby strengthening the success rate of green research and propelling the trajectory of green innovation.

It is undoubtedly acknowledged that sustainable development heavily relies on technological innovation. Green technologies reduce overall production costs, enhance production efficiency, establish consumer preference in the competitive market, and increase competitiveness. In addition, engaging in green innovation activities strengthens satisfaction levels among internal staff [57], external financial investors [28], and society. Under external scrutiny, firms are incentivized to uphold environmental responsibility, augment reputations, and elevate ESG performance [58]. Based on this, Hypothesis 4 is proposed in this study:

*Hypothesis 4*: Green funds can enhance corporate ESG performance by fostering green innovation.

## Section 3: Data and model

### 3.1 Variable selection

**3.1.1 Explained variable: Corporate ESG performance.** In this study, the chosen metric for measuring corporate ESG performance is *the Huazheng ESG Rating*, which has gained recognition and widespread application in both industry and academia. *The Huazheng ESG Rating* comprises nine levels, evaluated four times annually. From lowest to highest, they are denoted as C, CC, CCC, B, BB, BBB, A, AA, and AAA. Following established practices in existing literature [59], this study assigns values from 1 to 9 to the ratings C to AAA, respectively. The average score of each annual evaluation is adopted as the company's ESG performance for that year, where a higher score indicates superior ESG performance. Additionally, to ensure the robustness of our conclusions, we employ alternative ESG performance score data provided by the Bloomberg database for validation [60].

**3.1.2 Core explanatory variable: Green funds.** The proposed methodology to identify green funds in this study entails a two-step process. Initially, the "Fund Basic Information Table" and the "Stock Investment Detail Table" from the CSMAR database's fund market series are cross-referenced, yielding a detailed account of funds invested in Chinese listed enterprises. According to the provisions of the "Regulations on the Operation and Management of Publicly Offered Securities Investment Funds," a manual search is conducted to determine whether the "Investment Objectives" and "Investment Scope" of each fund include "environment-related" terms. If a fund displays environmentally relevant terms, it is considered as a green fund. Conversely, the absence of such terminology denotes other types of funds [61]. This research quantifies the number of green funds for a company in a specific year, which is then natural-log transformed with an added value of 1. The environmentally relevant terms are in **Appendix B in** S1 File.

**3.1.3 Mechanism variables.** Mechanisms for this study are selected based on three dimensions: financial constraints, corporate governance, and green innovation.

*Financial constraints*. SA Index (*SA*), FC Index (*FC*), Debt Financing Cost (*Cost*), Equity Capital Cost (*MPEG*), and Interfund Surplus (*InterFund*) are utilized in this study. SA and FC indices reflect the overall status of financing. Debt financing cost is computed as the

proportion of total interest expenses, fees, and other financial charges to the year-end total liabilities [10]. The MPEG model gauges equity financing costs [62]. The internal fund surplus is assessed using the ratio of operating cash flow to total assets at the beginning of the year.

*Corporate governance efficiency.* Agency Cost (*ATO*), Managerial myopia (*Myopia*), and Internal Control Quality (*lnIC*) are employed. Agency cost is measured by the total asset turnover rate [63]. The higher the total asset turnover rate, the lower the agency cost. Managerial myopia is quantified by the percentage of the total word frequency of terms indicating short-sighted behavior in corporate social responsibility reports [64]. The logarithm of the comprehensive internal control index disclosed in the DiBo database represents internal control quality.

*Green innovation.* Green patents (*lnGreTotal*) are categorized into green invention patents (*lnGreInv*) and green utility patents (*lnGreUti*) applied for the current year. Each number of green patents is subjected to a natural logarithm transformation with an added value of 1 [65].

**3.1.4 Control variables.** Drawing from related literature [18, 25, 60], we control for these factors: firm Size (*Size*), return on Assets (*ROA*), debt-to-asset ratio (*Lev*), firm Age (*FirmAge*), cash flow ratio (*Cashflow*), proportion of shares held by the largest shareholder (*TOP1*), proportion of independent directors (*Indep*) and board Size (*Board*). As outlined in Table 1, the descriptive statistics provide an overview of the variables in this research. The definition and source of all variables used in the empirical analysis are shown in **Appendix A in** S1 File.

As depicted in Table 1, the mean value of the dependent variable is 4.0525, with a standard deviation of 1.1116. The range spans from 1 to 8, revealing discernible variations in ESG ratings across diverse enterprises. The average ESG performance falls approximately within the range corresponding to a B grade, indicating that the ESG performance of most companies is positioned within the B-BBB interval. About the variable *green*, its mean stands at 0.4246, signifying that within the chosen sample, the proportion of companies associated with green funds reaches 42.46%. However, the average value for *lngreen* is merely 0.5595, underscoring the limited prevalence of green funds within the broader context. This suggests that the overall count of green funds remains modest, implying a restricted capacity for companies to attract substantial green funds. The distributions of the remaining control and instrumental variables align with those observed in existing literature, and no conspicuous outliers are detected.

## 3.2 Data processing methods and sources

This study focuses on the period from 2010 to 2021 and utilizes listed companies on the Shanghai and Shenzhen stock exchanges (A-shares) as the research sample. The following selection

**Table 1. Descriptive statistics of variables.**

| Variable | N | Mean | SD | Min | Max |
|---|---|---|---|---|---|
| ESG | 29,991 | 4.0525 | 1.1116 | 1.0000 | 8.0000 |
| ESG_pengbo | 10,580 | 28.1725 | 8.8798 | 11.7493 | 55.5980 |
| lngreen | 29,991 | 0.5595 | 0.7642 | 0.0000 | 2.8332 |
| green | 29,991 | 0.4246 | 0.4943 | 0.0000 | 1.0000 |
| greenratio | 29,991 | 0.0595 | 0.1564 | 0.0000 | 0.9819 |
| Size | 29,991 | 22.1749 | 1.2577 | 19.8898 | 26.0699 |
| Lev | 29,991 | 0.4217 | 0.2021 | 0.0539 | 0.8935 |
| ROA | 29,991 | 0.0405 | 0.0653 | -0.2402 | 0.2233 |
| FirmAge | 29,991 | 2.8820 | 0.3347 | 1.7918 | 3.4965 |
| Cashflow | 29,991 | 0.0477 | 0.0673 | -0.1501 | 0.2387 |
| TOP1 | 29,991 | 0.3397 | 0.1474 | 0.0850 | 0.7366 |
| Indep | 29,991 | 0.3757 | 0.0537 | 0.3333 | 0.5714 |
| Board | 29,991 | 2.1279 | 0.1992 | 1.6094 | 2.7081 |

criteria are employed: (1) Exclusion of samples in the financial and real estate sectors. (2) Exclusion of samples with abnormal trading statuses (ST or PT). (3) Exclusion of samples with missing variables. This process results in a final dataset with 29,991 observations. Furthermore, a winorization of 1% and 99% is carried out on all continuous variables. This method aims to mitigate the impact of outlier values on regression outcomes.

The ESG rating data for the companies under investigation are sourced from the Wind and Bloomberg database. Information about green funds was obtained from the CSMAR database. Other company-specific data are from the CSMAR, CNRDS, Dibo databases, Wingo platform, and official websites of fund companies.

## 3.3 Model setting

To empirically examine whether green funds influence corporate ESG performance, this study constructs the following baseline model:

$$ESG_{it} = \alpha_0 + \alpha_1 \ln green_{it} + \alpha_2 Controls_{it} + \mu_i + \delta_t + \varepsilon_{it} \tag{1}$$

where $ESG_{it}$ denotes the ESG performance of firm i in year t. $lngreen_{it}$ signifies the logarithm of the number of green funds. $Control_{it}$ encompasses firm-level control variables. $\mu_i$ captures firm-fixed effects, $\delta_t$ denotes time-fixed effects, and $\epsilon_{it}$ represents the error term. The coefficient $\alpha_1$ reflects the impact of green funds on corporate ESG performance. Consistent with hypothesis 1, it is expected that $\alpha_1$ will be positive, implying that a more significant presence of green funds is associated with improved corporate ESG performance.

## Section 4: Empirical result analysis

### 4.1 Validity testing of the selected indicators

Since this research identifies the business goals and scope of investment funds combined with textual analysis, the potential risk of "greenwashing" and other issues may still exist. Therefore, before conducting regression analysis, this study adopts various testing measures to verify the validity of selected indicators. Initially, we categorize the sample companies into polluting and clean industries based on their industrial attributes, grouping them according to the median of environmental performance indicators from the Huazheng ESG Rating. Subsequently, we observe the entry of green funds, presenting the results in Table 2 to demonstrate that the mean values of both the number of green funds (*greennumber*) and the proportion of market value held by green funds to the net value of the company (*greenratio*) are significantly higher in clean industries and firms with better environmental performance. It initially suggests that

**Table 2. Descriptive statistics of distinguishing among different categorized enterprises.**

| Classification Criteria | | *greennumber* | | | *greenratio* | | |
|---|---|---|---|---|---|---|---|
| | N | Mean | Min | Max | Mean | Min | Max |
| *Polluting Industry* | 7,179 | 1.2999 | 0 | 50 | 0.0437 | 0 | 0.9819 |
| *Clean Industry* | 22,812 | 1.6937 | 0 | 68 | 0.0644 | 0 | 0.9819 |
| Mean difference test (t-value) | | 8.7055*** | | | 9.7925*** | | |
| *Better Environmental Performance Enterprises* | 15,028 | 1.9327 | 0 | 68 | 0.0706 | 0 | 0.9819 |
| *Poor Environmental Performance Enterprises* | 14,963 | 1.2647 | 0 | 48 | 0.0483 | 0 | 0.9819 |
| Mean difference test (t-value) | | 17.3720*** | | | 12.3743*** | | |

Note:

*** indicates significance at the 1% level.

**Table 3. Corporate EID and entry of green funds.**

|  | (1) | (2) | (3) | (4) | (5) | (6) | (7) | (8) |
|---|---|---|---|---|---|---|---|---|
|  | *EPC* | *EG* | *EMS* | *EDT* | *EA* | *EEM* | *EHA* | *TS* |
|  | *lngreen* | *lngreen* | *lngreen* | *lngreen* | *lngreen* | *lngreen* | *lngreen* | *lngreen* |
| *EID* | 0.0314*** | 0.0180 | 0.0175* | 0.0307** | 0.0459*** | 0.0506*** | 0.0371*** | 0.0700*** |
|  | (0.0113) | (0.0143) | (0.0105) | (0.0156) | (0.0138) | (0.0139) | (0.0136) | (0.0190) |
| *Controls* | Yes | Yes | Yes | Yes | Yes | Yes | Yes | Yes |
| Firm fixed effects | Yes | Yes | Yes | Yes | Yes | Yes | Yes | Yes |
| Year fixed effects | Yes | Yes | Yes | Yes | Yes | Yes | Yes | Yes |
| N | 29,991 | 29,991 | 29,991 | 29,991 | 29,991 | 29,991 | 29,991 | 29,991 |
| Adj. $R^2$ | 0.5353 | 0.5352 | 0.5352 | 0.5352 | 0.5354 | 0.5355 | 0.5353 | 0.5356 |

Notes:

\*\*\*, \*\*, and \* indicate significance at the 1%, 5%, and 10% levels, respectively. The numbers in the parenthesis are robust standard errors clustered at the firm level.

green funds are more inclined to enter clean industries and firms with better environmental performance.

Furthermore, we argue our analysis by merging environmental information disclosure (*EID*) data of Chinese listed companies with our original database and examining whether the EID of companies attracts green funds to invest for further mitigating the potential risk of greenwashing. Specifically, EID contains eight dimensions of environmental information disclosure of listed companies, including environmental protection concept (*EPC*), environmental protection goal (*EG*), environmental management system (*EMS*), environmental protection education and training (*EDT*), environmental protection unique action (*EA*), environmental emergency mechanism (*EEM*), environmental protection honors or rewards (*EHA*), and three simultaneous systems (*TS*). According to the regression results in Table 3, the high-quality environmental information disclosure of enterprises improves the investment willingness of green funds, which indicates that the investment field of green funds also attaches great importance to the disclosure of the enterprise's environmental protection concept, environmental management system construction, environmental protection particular action and other essential measures. It indirectly confirms that the green funds we identified have a close relationship with the environmental protection behaviors of the enterprises. The analysis above confirms that green funds flow to enterprises and projects with green attributes and high-quality environmental information disclosure indeed, which reduces the possibility of "greenwashing".

## 4.2 Baseline regression results

The baseline regression results are presented in Table 4. The results without fixed effects or control variables are shown in Column (1). Column (2) includes control variables but without fixed effects. The results in Column (3) consider both control variables and fixed effects, indicating that a 1% increase in green funds corresponds to an average enhancement of 0.0655 units in corporate ESG performance, which substantiates Hypothesis 1 that an augmentation in the number of green funds significantly elevates corporate ESG performance. The entrance of green funds provides substantial financial resources for companies to undertake environmental initiatives and enhance governance efficiency. As a form of external monitoring, a higher density of green funds correlates with a more pronounced inclination for companies to

**Table 4. Baseline results.**

|  | (1) | (2) | (3) |
|---|---|---|---|
|  | ESG | ESG | ESG |
| lngreen | 0.2977*** | 0.1026*** | 0.0655*** |
|  | (0.0144) | (0.0141) | (0.0108) |
| Size |  | 0.2001*** | 0.2407*** |
|  |  | (0.0124) | (0.0205) |
| Lev |  | -0.8349*** | -0.9600*** |
|  |  | (0.0720) | (0.0759) |
| ROA |  | 2.6122*** | 0.9451*** |
|  |  | (0.1674) | (0.1399) |
| FirmAge |  | -0.1246*** | -0.3031* |
|  |  | (0.0361) | (0.1615) |
| Cashflow |  | -0.2538* | -0.4560*** |
|  |  | (0.1356) | (0.1034) |
| TOP1 |  | 0.2103** | 0.5003*** |
|  |  | (0.0881) | (0.1386) |
| Indep |  | 1.7116*** | 1.4491*** |
|  |  | (0.2430) | (0.2393) |
| Board |  | 0.2315*** | 0.1215 |
|  |  | (0.0725) | (0.0810) |
| _cons | 3.8859*** | -1.0327*** | -1.0323 |
|  | (0.0164) | (0.2908) | (0.6569) |
| Firm fixed effects | No | No | Yes |
| Year fixed effects | No | No | Yes |
| N | 29,991 | 29,991 | 29,991 |
| Adj. $R^2$ | 0.0419 | 0.1189 | 0.5130 |

engage in green governance. Consequently, the magnitude of improvement in corporate ESG performance is more substantial.

Regarding the control variables, larger firms, lower leverage ratios, superior operational performance, lower cash flow ratios, younger firms, higher ownership stakes by the largest shareholder, and a higher proportion of independent directors all improve ESG performance. These findings align with conclusions drawn by the majority of existing literature [18, 25, 60]. In addition, this study further analyses the differential impact of green funds on corporate environmental, social, and governance performance by splitting the ESG scores into three dimensions: E, S, and G. According to the regression results in Table 5, it is evident that green

**Table 5. The impact of three dimensions.**

|  | (1) | (2) | (3) |
|---|---|---|---|
|  | E_score | S_score | G_score |
| lngreen | 0.0655*** | 0.0209* | 0.0822*** |
|  | (0.0108) | (0.0111) | (0.0145) |
| Controls | Yes | Yes | Yes |
| Firm fixed effects | Yes | Yes | Yes |
| Year fixed effects | Yes | Yes | Yes |
| N | 29,991 | 29,991 | 29,991 |
| Adj. $R^2$ | 0.5130 | 0.6275 | 0.4633 |

**Table 6. The alternative measurements of key variables.**

|  | (1) | (2) | (3) |
|---|---|---|---|
|  | *ESG* | *ESG* | *ESG_pengbo* |
| *green* | 0.0516*** |  |  |
|  | (0.0136) |  |  |
| *greenratio* |  | 0.2446*** |  |
|  |  | (0.0498) |  |
| *lngreen* |  |  | 0.3495*** |
|  |  |  | (0.0940) |
| Controls | Yes | Yes | Yes |
| Firm fixed effects | Yes | Yes | Yes |
| Year fixed effects | Yes | Yes | Yes |
| N | 29,991 | 29,991 | 10,580 |
| Adj. R$^2$ | 0.5124 | 0.5127 | 0.8211 |

funds primarily enhance corporate environmental (*E_score*) and governance (*G_score*) performance. However, the effects of green funds on social responsibility performance (*S_score*) are relatively small. Therefore, green funds may be more inclined to focus on corporate environmental awareness and internal governance performance, and may not pay enough attention to promoting corporate social responsibility.

## 4.3 Robustness tests

**4.3.1 The replacement of key variables.**   In this study, a binary variable denoting the presence (1) or absence (0) of green funds entering a firm is utilized as a substitute for the count of green funds. Additionally, the proportion of market value held by green funds to the net value (*greenratio*) is employed as an alternative explanatory variable for regression analysis. As presented in Columns (1) and (2) of Table 6, firms with green funds experience an average increase of 0.0516 in ESG compared to firms without green funds. The coefficient of *greenratio* indicates that a 1% increase in green ratio results in an average enhancement of 0.2446 in ESG performance.

To ensure robustness, this study replaces the ESG rating data from HuaZheng with ESG performance scores sourced from Bloomberg's database for regression analysis (Due to the availability of Bloomberg ESG scores starting from 2011, the sample period for this robustness test is set from 2011 to 2021). As presented in Column (3) of Table 6, the results confirm the robustness of the conclusion that an increase in the number of green funds significantly enhances corporate ESG performance.

**4.3.2 The lagged effect.**   Recognizing that the effect of green funds on corporate ESG performance may extend beyond the current period, we further introduce a variable representing the number of green funds with a one-period lag (*L.lngreen*) into the regression model and lag all control variables by one-period (*L.Controls*). This extension allows us to examine the lagged effect of green funds on corporate ESG performance. The specific results are presented in Table 7. The coefficient of *L.lngreen* is significantly positive. However, the coefficient of *L.lngreen* (Column 2) is smaller than that of *lngreen* in the baseline model (Column 1). The lagged effect of green funds slightly decreases and remains significantly positive. It infers that the enhancement of ESG performance exhibits sustained positive considering lagged effect [66].

**Table 7. The lagged effect of green funds.**

|  | (1) | (2) |
|---|---|---|
|  | *ESG* | *ESG* |
| *lngreen* | 0.0655*** |  |
|  | (0.0108) |  |
| *L.lngreen* |  | 0.0463*** |
|  |  | (0.0117) |
| *Controls* | Yes | No |
| *L.Controls* | No | Yes |
| Firm fixed effects | Yes | Yes |
| Year fixed effects | Yes | Yes |
| N | 29,991 | 26,142 |
| Adj. R$^2$ | 0.5130 | 0.5446 |

### 4.3.3 Other important factors

Since the substantial impact of COVID-19 on both capital market and corporate decision-making, we exclude the samples from 2020 to 2021. The regression results in column (1) of Table 8 reveal that the estimated coefficient of *lngreen* remains significantly positive. Additionally, we further account for the potential influence of environmental regulations during the sample period by introducing low-carbon city pilot policy (*LCCP*), carbon market trading pilot policy (*Carbon*), and green credit policy (*GCP*) into the regression model. According to the estimation results in column (2) of Table 8, the fundamental conclusions of the study remain robust.

**Table 8. Other important factors.**

|  | (1) | (2) | (3) | (4) | (5) |
|---|---|---|---|---|---|
|  | Exclude the impact of COVID-19 | Control for environmental regulations | Exclude abnormal information disclosure samples | Keep high-quality information disclosure samples | Control for greenwashing of companies |
|  | *ESG* | *ESG* | *ESG* | *ESG* | *ESG* |
| *lngreen* | 0.0598*** | 0.0652*** | 0.0596*** | 0.0522*** | 0.0641*** |
|  | (0.0116) | (0.0108) | (0.0111) | (0.0106) | (0.0109) |
| *LCCP* |  | 0.0789** |  |  |  |
|  |  | (0.0305) |  |  |  |
| *Carbon* |  | 0.0211 |  |  |  |
|  |  | (0.0317) |  |  |  |
| *GCP* |  | 0.0302 |  |  |  |
|  |  | (0.0414) |  |  |  |
| *gws* |  |  |  |  | -0.0006 |
|  |  |  |  |  | (0.0250) |
| *Controls* | Yes | Yes | Yes | Yes | Yes |
| Firm fixed effects | Yes | Yes | Yes | Yes | Yes |
| Year fixed effects | Yes | Yes | Yes | Yes | Yes |
| N | 23,307 | 29,991 | 24,691 | 26,640 | 29,449 |
| Adj. R$^2$ | 0.5043 | 0.5132 | 0.5316 | 0.5174 | 0.5160 |

In light of the greenwashing phenomenon, we conduct a series of rigorous tests. Companies penalized by CSRC or stock exchanges during the sample period, notably due to issues like information disclosure, are excluded. The baseline results, as depicted in Column (3) of Table 8, retain their validity. What's more, our focus narrow to listed companies with excellent or good information disclosure evaluation results from the Shenzhen and Shanghai Stock Exchange. In Column (4) of Table 8, the baseline results persist. Following Hu et al. (2023) [67], we effectively add corporate greenwashing behavior (*gws*) in the baseline regression. We examine corporate greenwashing behavior by comparing firms' verbal green claims with their actual environmental performance. Firstly, we create a list of words related to environmental topics. Then, we calculate the frequency of these words appearing in the Management Discussion and Analysis (MD&A) section of annual reports for each company. We create a dummy variable called *Oral*. If the frequency exceeds the industry median for that period, we label it as 1; otherwise, it is labeled as 0. Similarly, another dummy variable, *Actual*, is assigned a value of 1 if the company suffers environmental punishment during the year; otherwise, it is assigned a value of 0. Consequently, if *Oral* = 1 and *Actual* = 1, the corporate greenwashing behavior is set to 1; otherwise, it is set to 0. As illustrated in Column (5) of Table 8, the baseline results endure.

**4.3.4 Alternative model specifications.** Considering that the ESG ratings from HuaZheng are non-negative integers, potential bias might arise in fixed-effects regressions. Following Chen and Lym's approach [68], we employ the negative binomial regression for robustness checks. As shown in Column (1) of Table 9, the coefficient of *lngreen* remains significantly positive, confirming the robustness of our conclusions.

Given that different industries and provinces face distinct external environments during the sample period, and macro factors like policy changes in different years might influence the ESG performance of various industries and firms, this study introduces industry-year and province-year interaction fixed effects in baseline regression. As presented in Columns (2)-(4) of Table 9, the results demonstrate the robustness of our findings.

**4.3.5 Placebo test.** This study conducts a placebo test to account for the potential influence of unobservable omitted variables. Drawing on the approach of Wang et al. [22], all observations of the variable *lngreen* for each observation in the sample dataset are extracted. These values are then randomly allocated to each observation, creating a variable denoted as *lngreen_false*. Subsequently, this *lngreen_false* variable replaces the actual variable *lngreen* in re-estimating the baseline model. This entire process is repeated 1000 times. Fig 3 depicts the estimated coefficients and density distribution of all *lngreen_false* variables.

**Table 9. Alternative model specifications.**

|  | (1) | (2) | (3) | (4) |
|---|---|---|---|---|
|  | *ESG* | *ESG* | *ESG* | *ESG* |
| *lngreen* | 0.0146*** | 0.0630*** | 0.0680*** | 0.0646*** |
|  | (0.0026) | (0.0110) | (0.0108) | (0.0110) |
| *Controls* | Yes | Yes | Yes | Yes |
| Firm fixed effects | Yes | Yes | Yes | Yes |
| Year fixed effects | Yes | Yes | Yes | Yes |
| Industry-Year fixed effects | No | Yes | No | Yes |
| Province-Year fixed effects | No | No | Yes | Yes |
| N | 29,991 | 29,991 | 29,991 | 29,991 |
| Adj. $R^2$ |  | 0.5260 | 0.5183 | 0.5307 |
| lnalpha | -44.6471 |  |  |  |

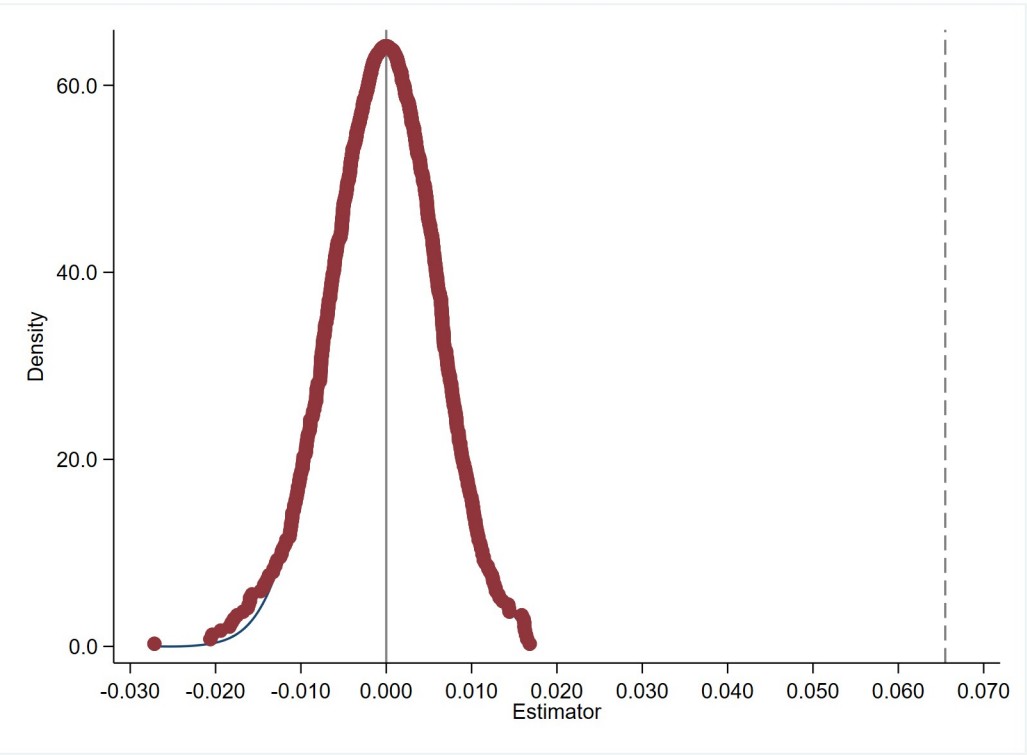

**Fig 3. Placebo test.**

Fig 3 shows that the estimated coefficients for "*lngreen_false*" are predominantly centered around zero, and they substantially deviate from the *lngreen* coefficients obtained in the baseline regression. This result demonstrates that unobserved variables are unlikely to affect the baseline regression results.

**4.3.6 Propensity score matching.**　Drawing inspiration from Huang et al. [69], this study employs propensity score matching (PSM) as an additional methodology to mitigate sample selection bias arising from differences in firm characteristics between firms with and without green funds. Initially, the study classifies firms into two categories based on the presence or absence of green funds. All control variables are utilized as covariates for matching. We apply the radius matching method, resulting in a matched sample of 29,851 observations. As presented in Table 10, the balance test results indicate that the t-tests for all covariates become insignificant after matching, suggesting the effectiveness of the PSM technique. Subsequently, regression analysis is performed using the matched sample, and the results are shown in Column (1) of Table 11. Additionally, this study uses 1:1 nearest-neighbor matching and kernel matching methods for matching firms with and without green funds. The regression results are shown in Columns (2) and (3) of Table 11. In all cases, the results verify the robustness of the baseline regression results.

## 4.4 Endogeneity test

**4.4.1 The instrumental variable approach.**　The enhancement of corporate ESG performance might not solely result from alleviating financial constraints and external monitoring prompted by green funds. It might be driven by the phenomenon that companies with better

**Table 10. PSM balanced test.**

| Variable | Unmatched | Mean | | | %reduct | t-test | |
|---|---|---|---|---|---|---|---|
| | Matched | Treated | Control | %bias | \|bias\| | t | p>\|t\| |
| Size | U | 22.633 | 21.837 | 65.6 | | 57.04 | 0.000 |
| | M | 22.619 | 22.606 | 1.0 | 98.4 | 0.78 | 0.438 |
| Lev | U | 0.422 | 0.421 | 0.6 | | 0.51 | 0.608 |
| | M | 0.423 | 0.421 | 0.7 | -17.2 | 0.57 | 0.571 |
| ROA | U | 0.061 | 0.025 | 58.0 | | 49.4 | 0.000 |
| | M | 0.061 | 0.061 | 0.2 | 99.7 | 0.14 | 0.889 |
| FirmAge | U | 2.854 | 2.903 | -14.8 | | -12.71 | 0.000 |
| | M | 2.854 | 2.850 | 1.3 | 91.0 | 1.00 | 0.316 |
| Cashflow | U | 0.060 | 0.039 | 31.1 | | 26.74 | 0.000 |
| | M | 0.059 | 0.060 | -1.0 | 96.7 | -0.81 | 0.420 |
| TOP1 | U | 0.345 | 0.336 | 6.2 | | 5.31 | 0.000 |
| | M | 0.345 | 0.342 | 2.0 | 68.3 | 1.52 | 0.129 |
| Indep | U | 0.376 | 0.376 | -0.1 | | -0.09 | 0.931 |
| | M | 0.376 | 0.375 | 1.3 | -1225.1 | 1.07 | 0.286 |
| Board | U | 2.146 | 2.114 | 16.1 | | 13.77 | 0.000 |
| | M | 2.146 | 2.144 | 1.0 | 93.6 | 0.81 | 0.418 |

ESG performance attract more green funds. Furthermore, unobservable factors influence both ESG performance and green funds. The presence of reverse causality and omitted variables may lead to endogeneity issues in this study. Therefore, we employ the interaction term consisting of whether the fund manager has a green background and the average green fund shareholding proportion of other firms in the same industry and the same year (excluding the focal firm) as an instrumental variable (*IV*). Among them, the background information data of the fund manager is manually crawled from the official website of the fund company. If the enterprise is held by at least one fund with a green background manager, the value is assigned to 1. The reason for this is that, on the one hand, the fund manager's green background influences the investment style and directs attention towards environment-friendly projects, consequently leading to investments in green funds. Therefore, this satisfies the relevance of *IV* (The green background of a fund manager encompasses both educational and working aspects. Green-related education is determined based on whether they majored in green-related specialities such as pulp and paper, environmental engineering, environmental science, and similar disciplines. Green work experience is assessed by examining their positions, specifically looking for roles within environmental departments, environmental protection agencies,

**Table 11. PSM results.**

| | (1) | (2) | (3) |
|---|---|---|---|
| | Radius | Nearest-neighbor | Kernel |
| lngreen | 0.0568*** | 0.0663*** | 0.0567*** |
| | (0.0132) | (0.0147) | (0.0131) |
| Controls | Yes | Yes | Yes |
| Firm fixed effects | Yes | Yes | Yes |
| Year fixed effects | Yes | Yes | Yes |
| N | 29,851 | 18,603 | 29,903 |
| Adj. $R^2$ | 0.5581 | 0.5411 | 0.5535 |

environmental committees, or pollution-control departments). On the other hand, the exogeneity of the instrument variable can be explained from two perspectives: Firstly, the green background of fund managers is usually driven by personal interests and future development goals, and is not directly related to corporate behaviors including ESG performance [70]. Secondly, the green background of fund managers is mostly determined in the early stage of their careers, and is not affected by the ESG performance of companies in the investment portfolio. This ensures that the green background is formed prior to investment decisions, thus satisfying the exclusion restrictions. This study employs a Two-Stage Least Squares (2SLS) approach to perform instrumental variable regression. The two-stage regression model is presented as Eqs (2) and (3):

$$\ln green_{it} = \beta_0 + \beta_1 IV_{it} + \beta_2 Controls_{it} + \varpi_i + \varpi_t + \varepsilon_{it}' \tag{2}$$

$$ESG_{it} = \alpha_0' + \alpha_1' \ln \widehat{green}_{it} + \alpha_2' Controls_{it} + \mu_i' + \delta_t' + \varepsilon_{it}'' \tag{3}$$

The results are presented in Table 12. Column (1) represents the first-stage regression outcome, indicating that *IV* positively relates to the number of green funds entering the focal company for the current year. The result of KleibergenPaap LM statistics rejects the assumption that the instrumental variable is unrecognized. The results of Cragg Donald Wald F and Kleibergen-Paap rk Wald F statistics reject that there is a weak instrument problem in this study. All the statistics are greater than the critical value at the 1% level. Hence, this instrumental variable is effective. Column (2) presents that the impact of green funds on corporate ESG performance is significantly positive, validating the robustness of the baseline regression results.

**4.4.2 Heckman two-stage model test.** The tendency of green funds to opt for companies with outstanding operational performance and sound governance structures might result in self-selection issues within the analysis sample of this study. To mitigate the potential estimation bias arising from this concern, this study draws inspiration from the approach of He et al. [71] and employs the Heckman two-stage model for examination. The first-stage probit regression model is presented as follows:

$$green_{it} = \beta_3 + \beta_4 Controls_{t-1} + \beta_5 SA_{t-1} + \beta_6 Indgreenfund_{t-1} + \beta_7 CPU_{t-1} + \varepsilon_{it-1}''' \tag{4}$$

**Table 12. IV estimations.**

|  | (1) | (2) |
|---|---|---|
|  | *lngreen* | *ESG* |
| *IV* | 11.9489*** |  |
|  | (1.5220) |  |
| *lngreen* |  | 0.4646** |
|  |  | (0.2017) |
| *Controls* | Yes | Yes |
| Firm fixed effects | Yes | Yes |
| Year fixed effects | Yes | Yes |
| N | 29,991 | 29,991 |
| Kleibergen-Paap rk LM statistic | 52.906 | |
| Kleibergen-Paap rk Wald F statistic | 61.635 | |
| Cragg-Donald Wald F statistic | 109.482 | |

We use whether a company has green funds in the current year as the dependent variable. The independent variables include the SA index which reflects financial constraints(*SA*), the industry-average green fund shareholding proportion excluding the focal firm(*Indgreenfund*), the climate policy uncertainty index for China which reflects regulation pressure(*CPU*), and other corporate characteristics [72–74]. We lag these independent variables by one period because green funds rely on past corporate operational performance and macro conditions to decide whether to invest in a listed company.

Financial constraints limit the ability to allocate resources towards sustainability initiatives, which makes it hard for companies to take full advantage of green funds. Therefore, green funds tend to invest in enterprises with less financial constraints [73]. Industry-level green fund preferences reflect strict sectoral requirements regarding environmental responsibility. Enterprises in industries with strong green preferences must improve their environmental performance in order to gain competitive advantages and win the favor of green funds. Regulation pressure influences corporate environmental behaviors by imposing legal mandates, incentives, or penalties related to environmental compliance [72, 74]. Therefore, enterprises accelerate green transformation to attract green funds. The business and governance conditions of companies are also important factors for the investment decisions of green fund investors. We calculate the corresponding inverse Mills ratio (*imr*) from the first-stage regression model and introduce it into the second-stage regression model. The second-stage regression model is presented as follows:

$$ESG_{it} = \lambda_0 + \lambda_1 \ln green_{it} + \lambda_2 Controls_{it} + \lambda_3 imr_{it} + \mu_i'' + \delta_i'' + \varepsilon_{it}'''' \tag{5}$$

where the inverse Mills ratio is incorporated into the baseline regression as an additional control variable to rectify the potential sample self-selection bias in the baseline regression.

The two-stage regression results are presented in Table 13, revealing that the *imr* is significantly negative with sample selection issues. The core explanatory variable remains significantly positive after adjusting for the *imr*, indicating that the baseline regression results are robust after addressing sample selectivity concerns.

## Section 5: Mechanism and heterogeneity analysis

### 5.1 Mechanism analysis

This section further examines the mechanisms from the perspectives of financial constraints, managerial efficiency, and green innovation.

**5.1.1 Financial constraints.** The presence of green funds is associated with lower financial constraints, resulting in increased funding availability for improving ESG performance. Green funds can enhance the ease of their financing processes, expand the range of financing options, and lower costs for debt and equity financing [31]. Strengthened financial resources empower companies to engage in green projects [54], undertake social responsibility [49], and reform internal governance structures.

In this study, five dimensions are employed to represent the level of corporate financial constraints, including the SA index, FC index, debt financing costs, equity financing costs, and internal funding adequacy. These dimensions are employed as dependent variables in the regression analysis presented in Table 14. The SA index and FC index reflect the overall financial constraints situation for companies; the larger their absolute values, the stronger the financial constraints. The results in Columns (1) and (2) demonstrate that green funds significantly reduce the financial constraints for companies. Debt financing and equity financing are two primary external financing channels for companies. According to Columns (3) and (4), more green funds are associated with lower debt and equity financing costs, thus lowering the

**Table 13. Heckman two-step estimations.**

| | (1) | | | (2) | |
|---|---|---|---|---|---|
| | First stage | | | Second stage | |
| Variables | *green* | | Variables | *ESG* | |
| L.Size | 0.4400*** | | lngreen | 0.0322*** | |
| | (0.0093) | | | (0.0115) | |
| L.Lev | -0.3912*** | | Size | 0.0724*** | |
| | (0.0539) | | | (0.0234) | |
| L.ROA | 6.0426*** | | Lev | -0.5052*** | |
| | (0.1868) | | | (0.0831) | |
| L.FirmAge | -0.0862 | | ROA | 0.9338*** | |
| | (0.0641) | | | (0.1422) | |
| L.Cashflow | 1.0373*** | | FirmAge | -0.3043* | |
| | (0.1409) | | | (0.1748) | |
| L.TOP1 | -0.7969*** | | Cashflow | -0.5198*** | |
| | (0.0599) | | | (0.1104) | |
| L.Indep | 0.3064 | | TOP1 | 0.5826*** | |
| | (0.1900) | | | (0.1371) | |
| L.Board | -0.1084* | | Indep | 1.4416*** | |
| | (0.0533) | | | (0.2447) | |
| L.SA | -0.4575*** | | Board | 0.0812 | |
| | (0.0889) | | | (0.0849) | |
| L.Indgreenfund | 50.2583*** | | imr | -0.5664*** | |
| | (2.6935) | | | (0.0250) | |
| L.CPU | -0.0035*** | | | | |
| | (0.0003) | | | | |
| Firm fixed effects | Yes | | | Yes | |
| Year fixed effects | Yes | | | Yes | |
| N | 26,142 | | | 26,142 | |
| Adj. $R^2$ | | | | 0.5452 | |

barriers to external funding for improving ESG performance. Column (5) demonstrates that green funds enhance a company's internal operating cash flow. Easing financial constraints can provide financial support for improving corporate ESG performance, which confirms hypothesis 2.

**5.1.2 Managerial efficiency.** With an increasing number of green funds in a company, the external supervisory role in corporate internal governance intensifies, thereby reducing

**Table 14. The mechanism results of financial constraints.**

| | (1) | (2) | (3) | (4) | (5) |
|---|---|---|---|---|---|
| | *SA* | *FC* | *Cost* | *MPEG* | *InterFund* |
| lngreen | -0.0056*** | -0.0217*** | -0.0007*** | -0.0012** | 0.0008*** |
| | (0.0010) | (0.0017) | (0.0001) | (0.0006) | (0.0003) |
| Controls | Yes | Yes | Yes | Yes | Yes |
| Firm fixed effects | Yes | Yes | Yes | Yes | Yes |
| Year fixed effects | Yes | Yes | Yes | Yes | Yes |
| N | 29,991 | 29,991 | 29,991 | 18,796 | 26,142 |
| Adj. $R^2$ | 0.9585 | 0.8036 | 0.6280 | 0.3860 | 0.9245 |

**Table 15. The mechanism results of managerial efficiency.**

|  | (1) | (2) | (3) |
|---|---|---|---|
|  | *ATO* | *Myopia* | *lnIC* |
| *lngreen* | 0.0106*** | -0.0026* | 0.0087*** |
|  | (0.0033) | (0.0013) | (0.0014) |
| *Controls* | Yes | Yes | Yes |
| Firm fixed effects | Yes | Yes | Yes |
| Year fixed effects | Yes | Yes | Yes |
| N | 29,991 | 29,991 | 28,759 |
| Adj. $R^2$ | 0.7942 | 0.4266 | 0.3111 |

agency costs, curbing short-sighted behaviors of management, and enhancing the quality of internal controls within the company. Improved corporate governance directs a company's attention towards sustainable development, leading to more green and socially responsible actions, consequently enhancing corporate ESG performance.

In this study, three indicators—total asset turnover, management myopia, and internal control index—are selected as dependent variables to analyze the mechanism of green funds on corporate governance. As presented in columns (1) and (3) of Table 15, a higher number of green funds is associated with higher total asset turnover and internal control quality. In Column (2) of Table 15, green funds effectively restrain opportunistic motives of management that pursue short-term gains at the expense of long-term welfare. In summary, an increased count of green funds enhances corporate information transparency and mitigates principal-agent problems between green fund investors and corporate managers. Under the external supervision of green fund investors, management's opportunistic behavior for short-term gains is effectively curtailed. It leads managers to focus on sustainable development, thereby enhancing internal control quality and alleviating agency problems. Improved managerial efficiency contributes to optimizing internal governance structures. Managers are more inclined to invest in green projects [75] and engage in sustainability practices [33]. It enables companies to simultaneously enhance their ESG performance and promote sustainable development. These results demonstrate the existence of hypothesis 3.

**5.1.3 Green innovation.** The green funds support the advancement of environmental projects and motivate companies to actively adopt green technologies. It is helpful to fulfill environmental responsibility, elevate public reputation, and enhance ESG performance.

In light of this, we employ green patents applied in the current year to examine the impact of green funds on corporate green innovation. What's more, we analyze the impact on green invention patents and green utility patents applied for the current year. The regression results are presented in Table 16. All results are significantly positive, indicating that green funds have

**Table 16. The mechanism results of green innovation.**

|  | (1) | (2) | (3) |
|---|---|---|---|
|  | *lnGreTotal* | *lnGreInvi* | *lnGreUti* |
| *lngreen* | 0.0207*** | 0.0189*** | 0.0160** |
|  | (0.0079) | (0.0066) | (0.0063) |
| *Controls* | Yes | Yes | Yes |
| Firm fixed effects | Yes | Yes | Yes |
| Year fixed effects | Yes | Yes | Yes |
| N | 29,991 | 29,991 | 29,991 |
| Adj. $R^2$ | 0.6621 | 0.6456 | 0.5974 |

a noteworthy positive effect on green patents. Green funds effectively stimulate the vitality of corporate green innovation. The green products and technologies developed through green innovation foster a more optimistic attitude towards the company's prospects among their internal staff [57], external financial investors [28], and consumers [65]. Under external incentives, companies engage in green and socially responsible activities more efficiently, thus providing a more decisive impetus for enhancing their ESG performance and convincing evidence of the correctness of hypothesis 4.

## 5.2 Heterogeneity analysis

This section conducts a heterogeneity analysis from three perspectives: external attention, firm characteristics, and regional features. Concerning external attention, this study examines the promoting impact of green funds on ESG performance across different sources of external attention from media, analysts, and the general public. Regarding firm characteristics, subsample tests are conducted based on ownership structure, pollution level, and firm size. At the regional level, sub-sample tests are performed based on environmental regulatory intensity, air pollution, and marketization.

**5.2.1 Heterogeneity analysis based on external attention.** As external supervisory roles, green funds effectively enhance corporate ESG performance. Enterprises are also subject to scrutiny from various external stakeholders, including the media, analysts, and the general public. External attention strengthens the promoting effect of green funds on corporate ESG performance. Therefore, this study examines the effect of green funds on corporate ESG performance across different levels of external attention from media, analysts, and the general public.

Media coverage guides enterprises better to concern their behaviors. The negative news reported by the media accentuates corporate reputational costs associated with environmental degradation, pollution, corruption, and dishonest business practices. Under the pressure of media coverage, companies opt for green governance to actively fulfill social responsibility and improve their ESG performance [39]. The frequency of company names in the titles of print and online news articles is logarithmically transformed to measure media attention. Columns (1) and (2) of Table 17 show grouped regressions conducted on high- and low-media attention groups based on the median. The results reveal that *lngreen* only remains positively significant in the high-media attention group. Permutation tests indicate a significant difference between the two groups, underscoring the more substantial influence of green funds on ESG performance in cases of greater media attention.

**Table 17. Heterogeneous analysis: External attention.**

|  | (1) | (2) | (3) | (4) | (5) | (6) |
|---|---|---|---|---|---|---|
|  | Media | | Analyst | | Public | |
|  | Low | High | Low | High | Low | High |
| *lngreen* | 0.0226 | 0.1009*** | 0.0217 | 0.0788*** | 0.0314 | 0.0836*** |
|  | (0.0172) | (0.0141) | (0.0178) | (0.0140) | (0.0194) | (0.0133) |
| *Controls* | Yes | Yes | Yes | Yes | Yes | Yes |
| Firm fixed effects | Yes | Yes | Yes | Yes | Yes | Yes |
| Year fixed effects | Yes | Yes | Yes | Yes | Yes | Yes |
| N | 14,874 | 15,117 | 11,513 | 18,478 | 10,825 | 19,166 |
| Adj. $R^2$ | 0.5486 | 0.5082 | 0.5037 | 0.5442 | 0.5163 | 0.5273 |
| Experience P-value | 0.000*** | | 0.000*** | | 0.000*** | |

Analysts gather internal information and disseminate research reports to external investors. It is beneficial to reducing information asymmetry and bolstering the supervisory roles of green funds over corporate management. Therefore, the management focuses more on sustainable development [41]. The number of analysts tracking a company plus one is logarithmically transformed to measure analyst attention. The sample is then grouped into high and low-analyst attention groups based on the median. Columns (3) and (4) indicate that *lngreen* is only positively significant at the 1% level in the high analyst attention group. Permutation tests show a significant difference between the two groups, highlighting that the positive impact of green funds on corporate ESG performance becomes more pronounced with more analyst attention.

Higher public environmental awareness encourages stronger public oversight of companies. Under the dual external supervision of green funds and the general public, companies face an increased severity of environmental penalties. It prompts companies to undertake green actions, optimize governance, and improve social reputation [76]. Public environmental awareness is measured by the annual average search volume of "environmental pollution" and "haze" on the Baidu search engine. The sample is grouped into high and low public environmental concern groups based on the median by merging the city-level public environmental concern data with firm data according to corporate registration information. The results in columns (5) and (6) show that the high public environmental concern group has a more pronounced estimate. Permutation tests indicate a significant difference between the two groups, underscoring that the promoting effect of green funds on corporate ESG performance is more pronounced in cases of more significant public environmental concern.

**5.2.2 Heterogeneity analysis based on firm characteristics.** From the ownership perspective, state-owned enterprises (SOEs) often benefit from government credit endorsements due to their extensive scale and government backing. It facilitates access to government financing, lowers financial constraints, and constructs well-established internal governance mechanisms. However, SOEs also bear more social responsibility due to stricter government oversight [77]. Conversely, non-SOE enterprises may face financing difficulties. Green funds enhance financing convenience and foster sustainable development practices. In Columns (1) and (2) of Table 18, green funds have a significant positive impact on the ESG performance of both SOEs and non-SOE enterprises, with the coefficient of *lngreen* being more significant in the non-SOE firms. Permutation tests indicate a significant difference between the two groups, suggesting that green funds play stronger roles in enhancing the ESG performance of non-SOE enterprises compared to SOEs.

**Table 18. Heterogeneous analysis: Firm characteristics.**

|  | (1) | (2) | (3) | (4) | (5) | (6) |
|---|---|---|---|---|---|---|
|  | Ownership | | Pollution | | Size | |
|  | SOE | non-SOE | Low | High | Small | Large |
| *lngreen* | 0.0371** | 0.0853*** | 0.0557*** | 0.1103*** | 0.0115 | 0.0817*** |
|  | (0.0170) | (0.0140) | (0.0120) | (0.0243) | (0.0164) | (0.0143) |
| *Controls* | Yes | Yes | Yes | Yes | Yes | Yes |
| Firm fixed effects | Yes | Yes | Yes | Yes | Yes | Yes |
| Year fixed effects | Yes | Yes | Yes | Yes | Yes | Yes |
| N | 10,892 | 19,099 | 22,812 | 7,179 | 14,995 | 14,996 |
| Adj. $R^2$ | 0.5423 | 0.5113 | 0.5254 | 0.4992 | 0.5267 | 0.5357 |
| Experience P-value | 0.000*** | | 0.000*** | | 0.000*** | |

Considering the nature of industry, enterprises in heavily polluting industries face stricter environmental regulations and higher financial constraints than those in less polluting industries [78]. Green funds provide sufficient funding for environmental investments and encourage green transformation, thereby boosting green innovation, internal governance efficiency, and corporate social responsibility. Therefore, the promoting effect of green funds on ESG performance may be more assertive in heavily polluting industries. Accordingly, the sample is divided based on whether companies belong to heavily polluting industries, and grouped regressions are performed as indicated in columns (3) and (4) of Table 18. The results suggest that the enhancing effect of green funds on ESG performance is larger in heavily polluting industries. Permutation tests indicate a significant difference between the two groups.

In terms of firm size, larger enterprises are subject to greater external scrutiny and more stringent government regulations due to their maturity. Reputation and economic losses stemming from pollution and neglect of social responsibility are more strict for larger companies. Due to their relatively robust governance systems, existing research indicates that large enterprises incur significantly lower costs in adhering to environmental regulations than small businesses [79]. However, the cost of pollution control is markedly higher for large enterprises than their smaller counterparts. Green fund investors impose strict oversight on the behavior of large enterprises. As a result, this study anticipates that the promoting effect of green funds is stronger in larger enterprises. To examine this fact, the sample is divided into large and small enterprise groups for grouped regressions based on the median of total assets, as presented in Table 18, columns (5) and (6). The coefficient of *lngreen* is higher in the large enterprise group, while insignificant in the small enterprise group. Permutation tests reveal a significant difference between the two groups, indicating that green funds have a more substantial promoting effect on the ESG performance of larger enterprises. This finding implies that the influence of green funds is more substantial in larger corporations, potentially leading to a reinforcing effect that the stronger entities become stronger, and the weaker ones relatively weaker. Consequently, this may contribute to an amplification of differences among enterprises, creating environmental sustainability gaps between more and less resourceful entities.

**5.2.3 Heterogeneity analysis based on regional characteristics.**   Government environmental regulations are crucial in curbing environmental degradation and driving enterprises towards green development. By employing incentive-based and mandatory measures, government-led environmental regulatory approaches complement the external oversight exerted by green funds on corporate environmental and social performance. The complementary effect between green funds and government environmental regulations suggests that the positive effect of green funds on corporate ESG performance is more substantial in regions with low environmental regulations. We use the frequency of environmental-related terms appearing in municipal government work reports to assess the environmental regulation intensity. The sample is divided into high and low groups based on the median of environmental regulation intensity. In columns (1) and (2) of Table 19, the results indicate a larger coefficient of *lngreen* in the low environmental regulation intensity group, showing that the effect of green funds on corporate ESG improvement is more potent in regions with less stringent environmental regulations. Permutation tests indicate a significant difference between the two groups.

The escalation of regional air pollution intensifies environmental uncertainty for enterprises and spurs increased public and governmental supervision of environmental protection. Enterprises in areas with severe air pollution face strong pressure from numerous external supervision, including green funds. Under this influence, enterprises adjust their business objectives and engage in ESG activities more actively. Consequently, in regions with higher levels of air pollution, the effect of green funds in enhancing corporate ESG performance is stronger. The study employs the annual average PM2.5 in municipalities to gauge the degree

**Table 19. Heterogeneous analysis: Region characteristics.**

| | (1) | (2) | (3) | (4) | (5) | (6) |
|---|---|---|---|---|---|---|
| | Environmental regulation | | Air pollution | | Marketability | |
| | Low | High | Low | High | Low | High |
| *lngreen* | 0.0777*** | 0.0516*** | 0.0473*** | 0.0660*** | 0.0795*** | 0.0619*** |
| | (0.0160) | (0.0141) | (0.0175) | (0.0138) | (0.0280) | (0.0118) |
| *Controls* | Yes | Yes | Yes | Yes | Yes | Yes |
| Firm fixed effects | Yes | Yes | Yes | Yes | Yes | Yes |
| Year fixed effects | Yes | Yes | Yes | Yes | Yes | Yes |
| N | 14,964 | 15,027 | 13,705 | 16,286 | 4,440 | 25,551 |
| Adj. $R^2$ | 0.5454 | 0.5154 | 0.5945 | 0.5098 | 0.5070 | 0.5120 |
| Experience P-value | 0.007*** | | 0.021** | | 0.000*** | |

of air pollution and categorizes companies into high and low-pollution groups based on the sample median by matching this data with enterprise registration information. The regression results, depicted in columns (3) and (4) of Table 19, reveal a higher coefficient of *lngreen* in the higher air pollution group with a significant difference according to the Permutation test. This result illustrates that the impact of green funds on corporate ESG improvement is more pronounced in regions with more severe air pollution.

In regions with inadequate institutional environments, companies face heightened information asymmetry and limited external scrutiny of their activities, weakening the impetus for companies to engage in ESG activities. Furthermore, acquiring external resources necessary for boosting ESG performance in regions with imperfect institutional environments becomes more challenging for companies. Under such circumstances, green funds can mitigate the deficiency in external oversight due to the lack of a sound institutional environment and provide the funding for ESG practices directly. Therefore, this study expects the effect of green funds on enhancing corporate ESG performance to be stronger in regions with imperfect institutional environments. The study selects the marketization index proposed by Fan Gang as a comprehensive measure of the regional institutional environment and matches it with the province information on enterprise registration. Using the median of the marketization index of 2010 as a standard, the sample is divided into high and low-marketization groups, followed by group regressions. The results in columns (5) and (6) of Table 19 reveal that compared to companies located in higher marketization regions, the effect of green funds on corporate ESG enhancement is more pronounced in companies situated in lower marketization regions. Permutation tests indicate a significant difference between the two groups.

## Section 6: Further analysis

While green funds can prompt companies to enhance their ESG performance, the question remains whether these green funds can further augment a company's production capacity and increase its economic value in the capital market. To address this question, this study assesses firms' total factor productivity using both the OP (*TFP_OP*) and LP (*TFP_LP*) methods, while market valuation is measured by Tobin's Q ratio (*TobinQ*) and the Price-to-Book ratio (*PB*). The regression results, as illustrated in Table 20, Columns (1) through (4), demonstrate that an increased presence of green funds corresponds to higher valuations in the capital market and improved production efficiency, with both results statistically significant. By comprehensively considering economic, social, and environmental impact during the investment process, green funds aim to maximize investment benefits. Consequently, invested companies usually exhibit

**Table 20. Further analysis.**

|  | (1) | (2) | (3) | (4) |
|---|---|---|---|---|
|  | *TobinQ* | *PB* | *TFP_LP* | *TFP_OP* |
| *lngreen* | 0.4383*** | 0.9312*** | 0.0240*** | 0.0214*** |
|  | (0.0150) | (0.0292) | (0.0049) | (0.0047) |
| *Controls* | Yes | Yes | Yes | Yes |
| Firm fixed effects | Yes | Yes | Yes | Yes |
| Year fixed effects | Yes | Yes | Yes | Yes |
| N | 29,460 | 29,404 | 29,548 | 29,548 |
| Adj. R$^2$ | 0.6516 | 0.6698 | 0.9119 | 0.8766 |

lower tendencies towards environmental violations and higher levels of internal governance efficiency [13]. With heightened efficiency in utilizing green funds, companies are more proactive in improving productivity and profitability. In conclusion, green funds not only advance corporate ESG performance but also elevate market valuation and production efficiency. Incorporating ESG factors in investment decisions optimizes resource allocation and facilitates economic prosperity.

## Section 7: Conclusion

This study empirically investigates the influence of green funds on corporate ESG performance using data from Chinese A-share listed companies. The results demonstrate that green funds improve corporate ESG performance. Mechanism analysis reveals that improved financing situations, enhanced managerial efficiency, and increased green innovation significantly contribute to the impact of green funds in promoting corporate ESG performance. Heterogeneity analysis claims that the positive effect is more pronounced in firms with higher external attention, non-state-owned enterprises, heavily polluting industries, and larger-scale companies. Additionally, the positive effect is more substantial in areas with lower environmental regulations, severe air pollution, and lower marketization. Furthermore, this study uncovers that green funds enhance corporate production capacity and economic value. Based on these findings, this study proposes the following policy recommendations:

Firstly, the government should promote the construction of a comprehensive green financial system to facilitate the flow of green funds to enterprises and reduce the threshold for green financing. For example, it should offer tax reductions or exemptions to individual and institutional investors showing a strong willingness to invest in green funds. What's more, the government can establish a dedicated fund to connect green funds with enterprises and provide financial subsidies or low-interest loans for green projects to lower the cost of environmental protection financing for enterprises, which can incentivize more capital to flow into green funds and promote more investment in environmental protection and sustainable development, thereby increasing the positive impact on corporate ESG performance. To enhance the inclusivity of green funds, the government can introduce a range of financing policies that lower the funding threshold for small-scale enterprises. Simultaneously, it should encourage fund investors to focus on the growth potential of these smaller enterprises, so that green funds can better empower their developments. Additionally, it is imperative to establish a specialized regulatory body tasked with overseeing the flow of green funds to avert the potential issue of "greenwashing." This regulatory department plays a crucial role in ensuring the legitimacy and authenticity of green investments, safeguarding the integrity of the overall green financial system.

Secondly, companies should recognize the supervisory effect exerted by external attention and ensure the efficient utilization of green funds. Ensuring the prudent use of green funds is paramount for maximizing their values. As crucial external stakeholders, green funds not only provide financial support for green development but also play vital roles in overseeing corporate performance. Consequently, companies should prioritize the strategic roles of external attention and ESG, actively engage in ESG practices, and enhance their capacities for sustainable development. Moreover, recognizing the paramount importance of transparency, enterprises should emphasize the disclosure of ESG issues. By improving the construction of the ESG information disclosure system, proactively assessing the quality of ESG information disclosure to regulatory authorities such as the China Securities Regulatory Commission (CSRC), and signaling commitments to sustainable development, companies can cultivate trust among stakeholders, thereby laying a solid foundation for realizing long-term value. However, it is also necessary to guard against issues such as excessive public opinion guidance or overheating of investor sentiment. Therefore, relevant departments need to monitor public opinions and capital market sentiment in real time to promote the rational investment of green funds.

Lastly, companies must embrace a pursuit of green innovation as a key driver for transformative change. This involves a robust commitment to intensifying R&D efforts geared towards the creation of green products and technologies. Such dedication serves to elevate a company's green reputation, making it an attractive prospect for investment from green funds. Specifically, companies should align their ESG governance practices with the preferences of green funds, actively introduce outstanding green innovative talents at home and abroad, and spearhead the large-scale adoption of green technologies. Ultimately, these concerted efforts will contribute to healthy and sustainable corporate growth. However, enterprises should also regularly assess green innovative talents, and adjust innovation direction according to the forefront of green knowledge and technologies to ensure the effectiveness of their own ESG construction.

Although this study only uses listed companies in China as the research sample, the conclusions are practically relevant for many countries with emerging economies, including China, for two main reasons. Firstly, most emerging economies have similar characteristics to China since their green financial systems are not yet fully developed and the concept of green sustainable development is yet to be promoted [80, 81]. Therefore, exploring how to guide sustainable development with the support of green funds is an integral part of green financial reform. Secondly, literature about green funds in different regions also provides support for the applicability of our conclusions. Ma et al. (2023) [61] discussed the relationship among green fund concerns, corporate R&D investment, and sustainable development. He et al. (2022) [71] and Jiang and Bai (2022) [82] analyzed the differential impacts of institutional investors on corporate green innovation. In addition, Siemroth and Hornuf (2023) [83] found that investors prefer positive environmental impacts and drive corporate transformation by investing in green projects. As for the performance of green funds in international markets, Goodell et al. (2022) [84] discovered that green funds are related to the development of financial technology. Ji et al. (2021) [81] revealed that the green funds of BRICS have performed better than other types of funds. Sangiorgi and Schopohl (2021) [12] and Chatnani (2018) [85] also found green investing funds in Europe and India have a better performance in stock price. Silva and Cortez (2016) [86] found that the performance of US and European green funds is higher in crisis periods. In general, our study is based on the above research and further focuses on the impact of green funds on the sustainable development of enterprises, rather than the single dimension of corporate environmental performance, stock price, or investment return considered in previous literature.

Our study extends the aforementioned research and delves deeper into examining green funds and corporate ESG performance. However, it is essential to note that this paper has yet to explicitly address the impact of small and medium-sized enterprises (SMEs), which remain pivotal to economic development in many emerging economies. Therefore, future research could be broadened to include SME data.

## Supporting information

**S1 File.**
(DOCX)

**S1 Data.**
(ZIP)

## Author Contributions

**Conceptualization:** Fangjun Wang, Tian Gan.

**Data curation:** Fangjun Wang.

**Formal analysis:** Fangjun Wang, Xinmiao Zhou.

**Investigation:** Fangjun Wang, Tian Gan.

**Methodology:** Fangjun Wang, Xinmiao Zhou.

**Project administration:** Fangjun Wang, Tian Gan.

**Supervision:** Xinmiao Zhou, Tian Gan.

**Validation:** Xinmiao Zhou, Tian Gan.

**Visualization:** Xinmiao Zhou.

**Writing – original draft:** Fangjun Wang.

**Writing – review & editing:** Xinmiao Zhou, Tian Gan.

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
