## [Decision Letter · Decision Letter 0]

27 Dec 2023

PONE-D-23-38405Can green funds improve corporate environmental, social, and governance performance? Evidence from Chinese-listed companiesPLOS ONE

Dear Dr. Gan,

Thank you for submitting your manuscript to PLOS ONE. After careful consideration, we feel that it has merit but does not fully meet PLOS ONE’s publication criteria as it currently stands. Therefore, we invite you to submit a revised version of the manuscript that addresses the points raised during the review process.

We look forward to receiving your revised manuscript.

Kind regards,

Khanh Hoang, Ph.D.

Academic Editor

PLOS ONE

 [This work was supported by China National Natural Science Foundation [71873096], Shanghai Planning Office of Philosophy and Social Science [2023EJB005], Major Supporting Project of the National Social Science Fund of China [21BJY002; 20AZD033], The Philosophy and Social Sciences in Zhejiang Province [20XXJC03ZD] and Humanities and Social Sciences Research Project of the Ministry of Education [20YJA790097].  

[This work was supported by China National Natural Science Foundation [71873096], Shanghai Planning Office of Philosophy and Social Science [2023EJB005], Major Supporting Project of the National Social Science Fund of China [21BJY002; 20AZD033], The Philosophy and Social Sciences in Zhejiang Province [20XXJC03ZD] and Humanities and Social Sciences Research Project of the Ministry of Education [20YJA790097].]

 [This work was supported by China National Natural Science Foundation [71873096], Shanghai Planning Office of Philosophy and Social Science [2023EJB005], Major Supporting Project of the National Social Science Fund of China [21BJY002; 20AZD033], The Philosophy and Social Sciences in Zhejiang Province [20XXJC03ZD] and Humanities and Social Sciences Research Project of the Ministry of Education [20YJA790097].  

5. We are unable to open your Supporting Information file [code.zip]. Please kindly revise as necessary and re-upload.

Reviewers' comments:

Reviewer's Responses to Questions

**Comments to the Author**

1. Is the manuscript technically sound, and do the data support the conclusions?

Reviewer #1: Yes

Reviewer #2: Partly

2. Has the statistical analysis been performed appropriately and rigorously? 

Reviewer #1: Yes

Reviewer #2: Yes

3. Have the authors made all data underlying the findings in their manuscript fully available?

Reviewer #1: Yes

Reviewer #2: Yes

4. Is the manuscript presented in an intelligible fashion and written in standard English?

Reviewer #1: Yes

Reviewer #2: Yes

5. Review Comments to the Author

Reviewer #1: While the paper provides valuable insights into the influence of green funds on ESG performance in Chinese A-share listed companies, there are a few areas where the authors could further develop and improve their work:

1. Causality and endogeneity: The authors should address the issue of endogeneity in their analysis. Green funds may be attracted to companies with better ESG performance, rather than being the cause of improved performance. To strengthen the causal claims made in the paper, the authors could consider employing instrumental variable techniques or conducting a difference-in-differences analysis.

2. Generalizability: While the study focuses on Chinese A-share listed companies, it would be helpful to discuss the generalizability of the findings to other contexts. Are there any unique characteristics of the Chinese market that may limit the applicability of the results to other countries or regions? Discussing the external validity of the findings would enhance the broader relevance of the study.

3. Counterarguments and limitations: It is important to acknowledge potential counterarguments and limitations of the study. For example, are there any potential negative effects or unintended consequences of green funds on ESG performance that should be considered? Addressing these points would provide a more balanced perspective and strengthen the overall argument.

4. Policy implications: While the authors provide some policy recommendations based on their findings, it would be helpful to elaborate on the practical implications of these recommendations. How feasible are these policy suggestions, and what challenges might arise in their implementation? Providing more detailed guidance for policymakers would enhance the practical relevance of the study.

By addressing these points, the authors can further strengthen the rigor, validity, and practical implications of their research.

Reviewer #2: The paper examines the effects of green funds and green innovation on ESG performance of publicly listed Chinese companies from 2010 to 2021. The research findings suggest that green funds can positively affect corporate ESG performance. Mechanism analysis indicates that green funds contribute to elevated ESG performance by mitigating financial constraints, enhancing managerial efficiency, and fostering green innovation. Heterogeneity analysis accentuates that the effect of green funds is particularly potent in companies with high external attention. Furthermore, the engagement of green funds also exerts a significant affirmative influence on its production capabilities and economic value. The paper is interesting, and I have some comments for the authors to improve the paper.

1. The sample period includes the COVID-19 lockdown period, which might cause some biases. The authors should control or isolate the potential impact of COVID-19 on their results.

2. It will also be interesting for the readers to see how the effects of green funds on ESG performance may vary due to national and international environmental regulations such as the Paris Climate Agreement.

3. The authors need to clarify how they control for greenwashing when they identify green funds. This is crucial as many funds may simply use environmentally relevant terms to promote their funds without actually being green (i.e. greenwashing).

4. The results must be discussed in light of the current literature to augment the contributions of the paper.

5. The paper requires proofreading to check for grammatical and spelling errors.

6. PLOS authors have the option to publish the peer review history of their article (what does this mean?). If published, this will include your full peer review and any attached files.

Reviewer #1: No

Reviewer #2: No

---

## [Author Response · Author response to Decision Letter 0]

27 Jan 2024

Reply to Reviewer 1

Dear Professor,

We sincerely thank you for your helpful and constructive comments and suggestions. We have considered your advice very seriously and incorporated it into the revised paper. We believe the paper is much improved and sincerely thank you for the advice and encouragement. Thank you for your kind comments on this manuscript. This statement details the changes we made, and it is keyed to your report. For ease of reading, we have included the original comments in bold font, followed by our detailed response in highlighted blue font. The reply to the comments is listed as follows. 

While the paper provides valuable insights into the influence of green funds on ESG performance in Chinese A-share listed companies, there are a few areas where the authors could further develop and improve their work:

[Comment 1] Causality and endogeneity: The authors should address the issue of endogeneity in their analysis. Green funds may be attracted to companies with better ESG performance, rather than being the cause of improved performance. To strengthen the causal claims made in the paper, the authors could consider employing instrumental variable techniques or conducting a difference-in-differences analysis.

Response: 

Thanks for your comments. As for the endogeneity test, we re-select a more exogenous instrumental variable for analysis. We employ the interaction term consisting of whether the fund manager has a green background and the average green fund shareholding proportion of other firms in the same industry and the same year (excluding the focal firm) as an instrumental variable (IV).

[Comment 2] Generalizability: While the study focuses on Chinese A-share listed companies, it would be helpful to discuss the generalizability of the findings to other contexts. Are there any unique characteristics of the Chinese market that may limit the applicability of the results to other countries or regions? Discussing the external validity of the findings would enhance the broader relevance of the study.

Response: 

Thank you for your insightful comments regarding the generalizability of our study. We appreciate your suggestion to discuss the external validity of our findings. Our results are applicable to other countries or regions.

We have made some supplements in the discussion and limitation section at the end of the article. 

[Comment 3] Counterarguments and limitations: It is important to acknowledge potential counterarguments and limitations of the study. For example, are there any potential negative effects or unintended consequences of green funds on ESG performance that should be considered? Addressing these points would provide a more balanced perspective and strengthen the overall argument.

Response: 

Thank you for your insightful feedback. We appreciate your suggestion to address potential counterarguments and limitations in our study, and we agree that acknowledging unintended consequences is crucial for a more comprehensive understanding. In the regression analysis, we discuss unintended consequences from the following two aspects: Firstly, we further investigate the differentiated impact of green funds on three dimensions of E, S, and G.

Secondly, we explore and discuss any adverse impacts that may arise from the influence of green funds, considering both large and small enterprises. This implies a discussion of how different company sizes might experience varying effects from green funds. The specific ideas are mainly:

These adjustments aim to explicitly address the potential negative effects, unintended consequences of green funds on ESG performance and the limitations of this study, providing a more balanced perspective and strengthening the overall argument of our study. 

[Comment 4] Policy implications: While the authors provide some policy recommendations based on their findings, it would be helpful to elaborate on the practical implications of these recommendations. How feasible are these policy suggestions, and what challenges might arise in their implementation? Providing more detailed guidance for policymakers would enhance the practical relevance of the study.

Response: 

Thanks for your comments. To ensure policy recommendations are more targeted and practical, we have rewritten this section from three perspectives: green financial system, external supervision, and green innovation. 

Reply to Reviewer 2

Dear Professor,

We sincerely thank you for your helpful and constructive comments and suggestions. We have considered your advice very seriously and incorporated it into the revised paper. We believe the paper is much improved and sincerely thank you for the advice and encouragement. Thank you for your kind comments on this manuscript. This statement details the changes we made, and it is keyed to your report. For ease of reading, we have included the original comments in bold font, followed by our detailed response in highlighted blue font. The reply to the comments is listed as follows. 

The paper examines the effects of green funds and green innovation on ESG performance of publicly listed Chinese companies from 2010 to 2021. The research findings suggest that green funds can positively affect corporate ESG performance. Mechanism analysis indicates that green funds contribute to elevated ESG performance by mitigating financial constraints, enhancing managerial efficiency, and fostering green innovation. Heterogeneity analysis accentuates that the effect of green funds is particularly potent in companies with high external attention. Furthermore, the engagement of green funds also exerts a significant affirmative influence on its production capabilities and economic value. The paper is interesting, and I have some comments for the authors to improve the paper.

[Comment 1] The sample period includes the COVID-19 lockdown period, which might cause some biases. The authors should control or isolate the potential impact of COVID-19 on their results.

Thanks for your comments. We have addressed this concern by conducting robustness tests, excluding samples from 2020 to 2021. The results remain consistent with our original findings. 

[Comment 2] It will also be interesting for the readers to see how the effects of green funds on ESG performance may vary due to national and international environmental regulations such as the Paris Climate Agreement.

Response: 

Thanks for your comments. In the robustness test, we take into account the overall impact of environmental regulations. We also consider the separate impact of each environmental regulation. The conclusions remain robust. 

We also conduct regressions grouped by regional environmental regulatory intensity to examine variations in the effectiveness under heterogeneous environmental regulatory conditions.

 [Comment 3] The authors need to clarify how they control for greenwashing when they identify green funds. This is crucial as many funds may simply use environmentally relevant terms to promote their funds without actually being green (i.e. greenwashing).

Response: 

Thank you for your valuable feedback. In addressing the potential issue of "greenwashing," our efforts involve several measures. Firstly, we have demonstrated that green funds exhibit a preference for entering clean industries and firms with superior environmental performance. The revised manuscript is as follows:

Secondly, we have excluded samples with abnormal information disclosure and kept high-quality information disclosure samples. The revised manuscript is as follows:

Lastly, we have incorporated corporate greenwashing as a control variable, and the conclusions drawn from these methodologies remain robust. 

[Comment 4] The results must be discussed in light of the current literature to augment the contributions of the paper. 

Response: 

Thank you for your constructive feedback. We appreciate your guidance on discussing the results in the context of the current literature to strengthen the contributions of our paper. In response to your suggestion, we have revised the manuscript to provide a more detailed discussion of how our study contributes to the existing literature. 

Our study makes significant contributions to the literature in three aspects. Firstly, we expand the research on the determinants of corporate ESG performance. This expansion of the research scope enhances our understanding of the multifaceted factors influencing corporate sustainable development. Secondly, while there is a substantial body of literature exploring the general impact of green finance on corporate performance, our study adds value by specifically examining the nuanced effects of green funds on corporate ESG performance. This focused exploration deepens our understanding of the intricate relationship between green finance and sustainable development. Lastly, our research explores the heterogeneous effects of green funds on corporate ESG performance from a micro-level perspective. It broadens the analysis to encompass how external scrutiny impacts corporate ESG performance.

What’s more, in the discussion section at the end of the article, we introduced supplementary comparisons of similar research scenarios from domestic and foreign literature to highlight the contributions of this article.

[Comment 5] The paper requires proofreading to check for grammatical and spelling errors. 

Response: 

We apologize for these types in the original manuscript. We have corrected them as you suggested. In addition, we have also carefully checked and corrected potential errors related to grammar, syntax, consistency, and language choice. Moreover, we have used a professional editorial service to help us polish the flow, style, and language in this revised manuscript.

---

## [Decision Letter · Decision Letter 1]

27 Feb 2024

PONE-D-23-38405R1Can green funds improve corporate environmental, social, and governance performance? Evidence from Chinese-listed companiesPLOS ONE

Dear Dr. Gan,

Thank you for submitting your manuscript to PLOS ONE. After careful consideration, we feel that it has merit but does not fully meet PLOS ONE’s publication criteria as it currently stands. Therefore, we invite you to submit a revised version of the manuscript that addresses the points raised during the review process.

We look forward to receiving your revised manuscript.

Kind regards,

Khanh Hoang, Ph.D.

Academic Editor

PLOS ONE

**Additional Editor Comments:**

I have read the reviewers' reports, which are positive about this submission. Then I read the revised submission again to see if any potential problem persists. I see that despite the reviewers did their jobs, I found some parts of the paper that need improvement to ensure sound empirical analysis and interpretation. I listed them as follows:

1. The control variables: Similar to the explanatory variable (in a robustness test), control variables should also be lagged by one period rather than at the same time point with the dependent variable. The authors may need to provide additional robustness test addressing this.

2. The Two-step Heckman selection estimation.

In this context, firms self-select themself into the control group or the treatment group, and such decision need to be addressed as it exhibits potential selection bias in the data. The use of the two-step Heckman selection model is reasonable. However, the Heckman selection model requires the modelling of the selection variables with at least one new variable that is not in the second stage. The modelling of the binary green fund choice in this revision includes the one-year lagged control variables, thus does not seem good determinants of the green fund choice. The authors also do not explain clearly the modelling of the green fund choice, and the factors that may determine the green fund choice. Factors that determine green fund choice may include financial constraints, the industry-average green fund choice (excluding the focal firm), and regulation pressure (may be measured by the climate policy uncertainty index for China by Lin and Zhao (2023)), then firm-level characteristics.

Lin, B. & Zhao, H. (2023). Tracking policy uncertainty under climate change. Resource Policy 83, 103699. https://doi.org/10.1016/j.resourpol.2023.103699

Other comments:

- Numbers should have comma separators, for example, the number of observations in the tables, "14,874" instead of "14874". Please scan the whole paper to revise this number format.

- The statement " on the other hand, the link between corporate ESG performance and the fund manager's previous green experience is fragile, satisfying the exogeneity of IV" (Section 4.4.1.) is exhausted. The authors need to revise this to provide a better rationale for why the IV is exogenous and satisfies the exclusion restrictions.

- The definition of the gws variable is missing.

Reviewers' comments:

Reviewer's Responses to Questions

**Comments to the Author**

1. If the authors have adequately addressed your comments raised in a previous round of review and you feel that this manuscript is now acceptable for publication, you may indicate that here to bypass the “Comments to the Author” section, enter your conflict of interest statement in the “Confidential to Editor” section, and submit your "Accept" recommendation.

Reviewer #1: All comments have been addressed

Reviewer #2: All comments have been addressed

2. Is the manuscript technically sound, and do the data support the conclusions?

Reviewer #1: Yes

Reviewer #2: Yes

3. Has the statistical analysis been performed appropriately and rigorously? 

Reviewer #1: Yes

Reviewer #2: Yes

4. Have the authors made all data underlying the findings in their manuscript fully available?

Reviewer #1: Yes

Reviewer #2: Yes

5. Is the manuscript presented in an intelligible fashion and written in standard English?

Reviewer #1: Yes

Reviewer #2: Yes

6. Review Comments to the Author

Reviewer #1: Your research on the positive influence of green funds on corporate environmental, social, and governance (ESG) performance, as well as the underlying mechanisms driving this impact, provides valuable insights into sustainable development practices within the corporate sector. The systematic analysis of how green funds contribute to elevated ESG performance by alleviating financial constraints, enhancing managerial efficiency, and fostering green innovation is both thorough and compelling. Upon careful revision of your manuscript, I am assured that your research constitutes a substantial contribution to the academic field. I am of the opinion that the insights derived from your study will captivate our readership and foster a constructive dialogue within the realm of sustainable development and corporate sustainability.

Reviewer #2: Thank you for carefully addressing my comments. There is a minor point that can be addressed during the proof process. In Section 7, you should change the heading to Conclusion instead of Conclusion and Discussion.

7. PLOS authors have the option to publish the peer review history of their article (what does this mean?). If published, this will include your full peer review and any attached files.

Reviewer #1: No

Reviewer #2: No

---

## [Author Response · Author response to Decision Letter 1]

7 Mar 2024

Reply to Editors

Dear editors,

We sincerely thank you for your helpful and constructive comments and suggestions. We have considered your advice very seriously and incorporated it into the revised paper. We believe the paper is much improved and sincerely thank you for the advice and encouragement. Thank you for your kind comments on this manuscript. This statement details the changes we made, and it is keyed to your report. For ease of reading, we have included the original comments in bold font, followed by our detailed response in highlighted blue font. The reply to the comments is listed as follows. 

[Comment 1] The control variables: Similar to the explanatory variable (in a robustness test), control variables should also be lagged by one period rather than at the same time point with the dependent variable. The authors may need to provide additional robustness test addressing this.

Response: 

Thank you for your valuable suggestion. In response to your comment regarding the timing of control variables in our robustness tests, we have revised our analysis to include control variables that are lagged by one period. The revised manuscript is as follows:

Recognizing that the effect of green funds on corporate ESG performance may extend beyond the current period, we further introduce a variable representing the number of green funds with a one-period lag (L.lngreen) into the regression model and lag all control variables by one-period (L.Controls). This extension allows us to examine the lagged effect of green funds on corporate ESG performance. The specific results are presented in Table 7. The coefficient of L.lngreen is significantly positive. However, the coefficient of L.lngreen (Column 2) is smaller than that of lngreen in the baseline model (Column 1). The lagged effect of green funds slightly decreases and remains significantly positive. It infers that the enhancement of ESG performance exhibits sustained positive considering lagged effect [66].

Table 7. The lagged effect of green funds

[Comment 2] The Two-step Heckman selection estimation.

In this context, firms self-select themself into the control group or the treatment group, and such decision need to be addressed as it exhibits potential selection bias in the data. The use of the two-step Heckman selection model is reasonable. However, the Heckman selection model requires the modelling of the selection variables with at least one new variable that is not in the second stage. The modelling of the binary green fund choice in this revision includes the one-year lagged control variables, thus does not seem good determinants of the green fund choice. The authors also do not explain clearly the modelling of the green fund choice, and the factors that may determine the green fund choice. Factors that determine green fund choice may include financial constraints, the industry-average green fund choice (excluding the focal firm), and regulation pressure (may be measured by the climate policy uncertainty index for China by Lin and Zhao (2023)), then firm-level characteristics.

Lin, B. & Zhao, H. (2023). Tracking policy uncertainty under climate change. Resource Policy 83, 103699. https://doi.org/10.1016/j.resourpol.2023.103699

Response: 

Thank you for your insightful comments regarding the two-step Heckman selection model in our study. We acknowledge the importance of addressing potential selection bias as firms self-select into the control or treatment group regarding green fund choices. In response to your feedback, we have revised our Heckman two-stage model analysis to include financial constraints, the industry-average green fund choice (excluding the focal firm), regulatory pressure, and other firm-level characteristics in our model. The revised manuscript is as follows:

The tendency of green funds to opt for companies with outstanding operational performance and sound governance structures might result in self-selection issues within the analysis sample of this study. To mitigate the potential estimation bias arising from this concern, this study draws inspiration from the approach of He et al. [71] and employs the Heckman two-stage model for examination. The first-stage probit regression model is presented as follows:

We use whether a company has green funds in the current year as the dependent variable. The independent variables include the SA index which reflects financial constraints(SA), the industry-average green fund shareholding proportion excluding the focal firm(Indgreenfund), the climate policy uncertainty index for China which reflects regulation pressure(CPU), and other corporate characteristics [72–74]. We lag these independent variables by one period because green funds rely on past corporate operational performance and macro conditions to decide whether to invest in a listed company. 

Financial constraints limit the ability to allocate resources towards sustainability initiatives, which makes it hard for companies to take full advantage of green funds. Therefore, green funds tend to invest in enterprises with less financial constraints [73]. Industry-level green fund preferences reflect strict sectoral requirements regarding environmental responsibility. Enterprises in industries with strong green preferences must improve their environmental performance in order to gain competitive advantages and win the favor of green funds. Regulation pressure influences corporate environmental behaviors by imposing legal mandates, incentives, or penalties related to environmental compliance [72,74]. Therefore, enterprises accelerate green transformation to attract green funds. The business and governance conditions of companies are also important factors for the investment decisions of green fund investors. We calculate the corresponding inverse Mills ratio (imr) from the first-stage regression model and introduce it into the second-stage regression model. The second-stage regression model is presented as follows:

where the inverse Mills ratio is incorporated into the baseline regression as an additional control variable to rectify the potential sample self-selection bias in the baseline regression. 

Table 13. Heckman two-step estimations

The two-stage regression results are presented in Table 13, revealing that the imr is significantly negative with sample selection issues. The core explanatory variable remains significantly positive after adjusting for the imr, indicating that the baseline regression results are robust after addressing sample selectivity concerns.

Other comments:

[Comment 1] "14,874" instead of "14874". Please scan the whole paper to revise this number format.

Response: 

Thank you for pointing out the formatting inconsistency. We have carefully scanned the entire document and revised all instances of number formatting to ensure consistency.

[Comment 2] The statement " on the other hand, the link between corporate ESG performance and the fund manager's previous green experience is fragile, satisfying the exogeneity of IV" (Section 4.4.1.) is exhausted. The authors need to revise this to provide a better rationale for why the IV is exogenous and satisfies the exclusion restrictions.

Response: 

Thank you for your feedback. we have revised the instrumental variable section of our manuscript to provide a more comprehensive rationale for its exogeneity. The revised manuscript is as follows:

On the other hand, the exogeneity of the instrument variable can be explained from two perspectives: Firstly, the green background of fund managers is usually driven by personal interests and future development goals, and is not directly related to corporate behaviors including ESG performance [70]. Secondly, the green background of fund managers is mostly determined in the early stage of their careers, and is not affected by the ESG performance of companies in the investment portfolio. This ensures that the green background is formed prior to investment decisions, thus satisfying the exclusion restrictions.

[Comment 3] The definition of the gws variable is missing.

Response: 

Thank you for your comments. We apologize for the omission of the definition for the greenwashing behavior variable (gws) in the original manuscript. We have revised our manuscript to include a definition of the greenwashing behavior variable(gws). The revised manuscript is as follows:

Following Hu et al.(2023) [67], we effectively add corporate greenwashing behavior (gws) in the baseline regression. We examine corporate greenwashing behavior by comparing firms' verbal green claims with their actual environmental performance. Firstly, we create a list of words related to environmental topics. Then, we calculate the frequency of these words appearing in the Management Discussion and Analysis (MD&A) section of annual reports for each company. We create a dummy variable called Oral. If the frequency exceeds the industry median for that period, we label it as 1; otherwise, it is labeled as 0. Similarly, another dummy variable, Actual, is assigned a value of 1 if the company suffers environmental punishment during the year; otherwise, it is assigned a value of 0. Consequently, if Oral=1 and Actual=1, the corporate greenwashing behavior is set to 1; otherwise, it is set to 0.

Reference

Gan T, Li Y, Jiang Y. The impact of air pollution on venture capital: evidence from China. Environ Sci Pollut R. 2022;29(60):90615–31. doi: 10.1007/s11356-022-21972-7

Hu X, Hua R, Liu Q, Wang C. The green fog: Environmental rating disagreement and corporate greenwashing. Pacific-Basin Finance Journal. 2023;78:101952. doi: 10.1016/j.pacfin.2023.101952

Martin PR, Moser DV. Managers’ green investment disclosures and investors’ reaction. J Account Econ. 2016;61(1):239–54. doi: 10.1016/j.jacceco.2015.08.004

He F, Yan Y, Hao J, Wu J (George). Retail investor attention and corporate green innovation: Evidence from China. Energy Economics. 2022;115:106308. doi: 10.1016/j.eneco.2022.106308

Ren X, Zhang X, Yan C, Gozgor G. Climate policy uncertainty and firm-level total factor productivity: Evidence from China. Energ Econ. 2022;113:106209. doi: 10.1016/j.eneco.2022.106209

Cecere G, Corrocher N, Mancusi ML. Financial constraints and public funding of eco-innovation: empirical evidence from European SMEs. Small Bus Econ. 2020;54(1):285–302. doi: 10.1007/s11187-018-0090-9

Lin B, Zhao H. Tracking policy uncertainty under climate change. Resour Policy. 2023;83:103699. doi: 10.1016/j.resourpol.2023.103699

Reply to Reviewers

[Reviewer1] Your research on the positive influence of green funds on corporate environmental, social, and governance (ESG) performance, as well as the underlying mechanisms driving this impact, provides valuable insights into sustainable development practices within the corporate sector. The systematic analysis of how green funds contribute to elevated ESG performance by alleviating financial constraints, enhancing managerial efficiency, and fostering green innovation is both thorough and compelling. Upon careful revision of your manuscript, I am assured that your research constitutes a substantial contribution to the academic field. I am of the opinion that the insights derived from your study will captivate our readership and foster a constructive dialogue within the realm of sustainable development and corporate sustainability.

Response: 

Thank you for your encouraging feedback and constructive review of our manuscript. We are inspired by your supportive and insightful comments. We will strive for further research in the field of sustainable development.

[Reviewer2] Thank you for carefully addressing my comments. There is a minor point that can be addressed during the proof process. In Section 7, you should change the heading to Conclusion instead of Conclusion and Discussion.

Response: 

Thank you for your comments. We change the heading to Conclusion in Section 7.

---

## [Editor Report · Decision Letter 2]

14 Mar 2024

Can green funds improve corporate environmental, social, and governance performance? Evidence from Chinese-listed companies

PONE-D-23-38405R2

Dear Dr. Gan,

We’re pleased to inform you that your manuscript has been judged scientifically suitable for publication and will be formally accepted for publication once it meets all outstanding technical requirements.

Kind regards,

Khanh Hoang, Ph.D.

Academic Editor

PLOS ONE
---

## [Editor Report · Acceptance letter]

20 Mar 2024

PONE-D-23-38405R2 

PLOS ONE

Dear Dr. Gan, 

I'm pleased to inform you that your manuscript has been deemed suitable for publication in PLOS ONE. Congratulations! Your manuscript is now being handed over to our production team.

Kind regards, 

on behalf of

Dr Khanh Hoang 

Academic Editor

PLOS ONE